# TranX-Adapter: Bridging Artifacts and Semantics within MLLMs for Robust AI-generated Image Detection

Wenbin Wang [1*]   Yuge Huang [2]   Jianqing Xu [2]   Yue Yu [2]   Jiangtao Yan [2]
Shouhong Ding [2]   Pan Zhou [3]   Yong Luo [1]

## Abstract

Rapid advances in AI-generated image (AIGI) technology enable highly realistic synthesis, threatening public information integrity and security. Recent studies have demonstrated that incorporating texture-level artifact features alongside semantic features into multimodal large language models (MLLMs) can enhance their AIGI detection capability. However, our preliminary analyses reveal that artifact features exhibit high intra-feature similarity, leading to an almost uniform attention map after the softmax operation. This phenomenon causes `attention dilution`, thereby hindering effective fusion between semantic and artifact features. To overcome this limitation, we propose a lightweight fusion adapter, *TranX-Adapter*, which integrates a *Task-aware Optimal-Transport Fusion* that leverages the Jensen-Shannon divergence between artifact and semantic prediction probabilities as a cost matrix to transfer artifact information into semantic features, and an *X-Fusion* that employs cross-attention to transfer semantic information into artifact features. Experiments on standard AIGI detection benchmarks upon several advanced MLLMs, show that our *TranX-Adapter* brings consistent and significant improvements (up to +6% accuracy). Code is available at https://github.com/DreamMr/TranX-Adapter.

*This work was done during an internship at Tencent YouTu Lab. [1]School of Computer Science, National Engineering Research Center for Multimedia Software and Hubei Key Laboratory of Multimedia and Network Communication Engineering, Wuhan University [2]Tencent YouTu Lab [3]Singapore Management University. Correspondence to: Pan Zhou <panzhou@smu.edu.sg>, Yong Luo <luoyong@whu.edu.cn>.

*Proceedings of the 43rd International Conference on Machine Learning*, Seoul, South Korea. PMLR 306, 2026. Copyright 2026 by the author(s).

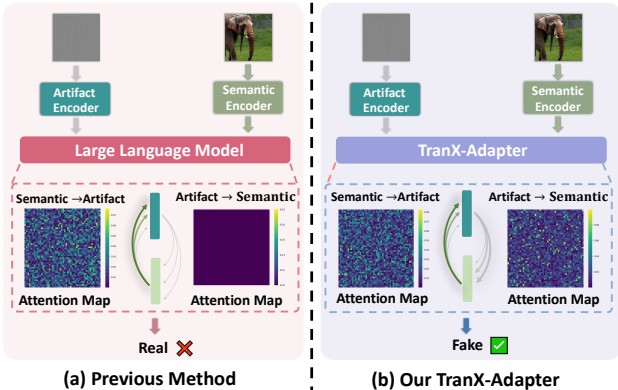

*Figure 1.* **Comparison between the previous fusion method and our *TranX-Adapter*.** (a) Previous Method: Concatenates artifact (*e.g.*, from NPR) and semantic features (*e.g.*, from CLIP-ViT), resulting in uniform attention and weak interaction. (b) Our *TranX-Adapter*: Incorporates a lightweight bidirectional fusion mechanism that enhances feature interaction.

## 1. Introduction

Recent progress in generative AI (Karras et al., 2019; Dhariwal & Nichol, 2021; Goodfellow et al., 2014; Rombach et al., 2022; Park et al., 2019; Zhu et al., 2017) has dramatically enhanced the capability to synthesize highly realistic images, enabling the creation of visual content that closely mimics real-world scenes. Despite these remarkable achievements, such technologies also pose substantial societal risks, as convincingly fake images can be exploited to mislead the public and propagate misinformation. These emerging threats have consequently driven the computer vision community to advance research in AI-generated image detection (AIGI detection) (Tan et al., 2024; Wang et al., 2020; Zhang et al., 2019; Qian et al., 2020; Liu et al., 2020; Ojha et al., 2023; Yan et al., 2025; Chen et al., 2025), aiming to develop more robust and reliable techniques for distinguishing synthetic image from authentic image.

Despite substantial progress in AIGI detection, notable challenges persist. Early work (Wang et al., 2020; Zhang et al., 2019; Tan et al., 2024) predominantly targets GAN and diffusion-generated images (Goodfellow et al., 2014; Dhariwal & Nichol, 2021), limiting robustness against the latest

generative models (Labs et al., 2025). To improve generalization, recent studies advance three complementary directions: (1) artifact-based methods (Tan et al., 2024; Qian et al., 2020; Zhong et al., 2023b; Yan et al., 2025), which capture pixel-level artifacts induced by up-sampling or interpolation; (2) semantic-based methods (Tan et al., 2025; Yan et al.; Xu et al., 2024), which leverage high-level visual semantics and broad world knowledge of Multimodal Large Language Models (MLLMs) to detect subtle or localized manipulations; and (3) hybrid artifact-semantic methods (Zhou et al., 2025; Cheng et al., 2025), the most promising direction, explicitly combine artifact encoders with MLLMs to combine precise pixel-level cues with semantic robustness. A representative example is AIGI-Holmes (Zhou et al., 2025), which integrates NPR (Tan et al., 2024) with LLaVA-1.6-mistral (Liu et al., 2024), yielding state-of-the-art (SOTA) performance across diverse AIGI detection benchmarks.

However, our preliminary analyses reveal that naively concatenating artifact features (*e.g.*, from NPR) with semantic features (*e.g.*, from CLIP-ViT) and feeding them into the Large Language Model (LLM) yields suboptimal fusion behavior. As illustrated in Figure 1 (a), when artifact features are used as *key* and *value* and semantic features as *query* to transfer artifact information into the semantic space (Artifact → Semantic), the resulting attention map collapses into an almost uniform pattern, manifesting the `attention dilution` phenomenon (Zhang et al., 2024). This degradation stems from the high intra-feature similarity of artifact representations, which suppresses discriminative cross-encoder interactions and limits the model's ability to effectively convey fine-grained, texture-level artifact cues.

To achieve more effective fusion between semantic and artifact features, we introduce a lightweight fusion adapter, termed ***TranX-Adapter***, which is placed before the LLM and employs distinct fusion strategies for different interaction directions. The proposed ***TranX-Adapter*** comprises two modules: *Task-Aware Optimal-Transport Fusion (TOP-Fusion)* and *X-Fusion*. Specifically, *TOP-Fusion* transfers artifact features into the semantic features by optimal transport (Cuturi, 2013). We employ the distance between the task-specific prediction probabilities of the two features, instead of relying on the dot-product interaction used in self-attention (Vaswani et al., 2017). This design mitigates the `attention dilution` induced by the high intra-feature similarity among artifact features. In contrast, *X-Fusion* transfer the semantic features into the artifact features by cross-attention. Here, we introduce *X-Fusion* based on the observation that the interaction between the two features predominantly emerges in the shallow layers of the LLM. Accordingly, we confine the trainable parameters to a lightweight module, avoiding any modification to the LLM and improving training efficiency.

Our contribution is identifying that fusing artifact and semantic features within MLLMs is hindered by the high intra-feature similarity of artifact representations, which weakens discriminative interactions. To address this challenge, we propose ***TranX-Adapter***, a lightweight adapter that enables effective bidirectional fusion through *TOP-Fusion* (Artifact → Semantic) and *X-Fusion* (Semantic → Artifact). Extensive experiments show that ***TranX-Adapter*** consistently improves performance across both cross-method and cross-dataset benchmarks, achieving an average gain of 4.7% and outperforming recent SOTA approaches.

## 2. Related Work

With the rapid progress of AI-based image generation, a broad spectrum of detectors has emerged. Recent studies advance three complementary directions: **(1) artifact-based methods**, **(2) semantic-based methods** and **(3) hybrid artifact-semantic methods.**

**Artifact-Based Methods.** Artifact-Based methods primarily aim to capture texture-level features induced by the up-sampling operations inherent in image generation models (Tan et al., 2024; Zhong et al., 2023a; Qian et al., 2020; Frank et al., 2020). These features manifest as distinctive pixel-level artifacts, such as similar values between adjacent pixels. Tan et al. (2024) propose NPR to capture localized structural cues introduced by the up-sampling operations in CNN-based generative networks. Zhong et al. (2023a) introduce PatchCraft, a texture-patch-based method that suppresses global semantics and exploits inter-pixel correlation contrasts to improve generalization.

**Semantic-Based Methods.** These methods identify synthetic images through semantic cues, such as human hand contours, by training deep neural networks on extensive real and generated datasets covering diverse object categories (Xu et al., 2024; Chang et al., 2023; Tan et al., 2025; Chen et al., 2024). A representative examples of these approaches are to leverage a powerful foundation model (*e.g.*, MLLMs) that possesses extensive world knowledge, thereby endowing it with strong generalization capability. Xu et al. (2024) present FakeShield that leverages LLM to not only detect image forgery and localize tampered regions but also produce human-readable rationales for its judgments. Chang et al. (2023) formulate deepfake image detection as a visual question-answering task and leverages prompt-tuned MLLM to significantly improve generalisability to unseen generative models.

**Hybrid Artifact-Semantic Methods.** These methods simultaneously integrate the previous two approaches by combining pixel-level artifact cues with semantic world knowledge, achieving notable performance gains (Cheng et al., 2025; Zhou et al., 2025). Cheng et al. (2025) introduce a

hybrid method CO-SPY that jointly enhances and adaptively fuses semantic and pixel-artifact features to robustly detect synthetic images across diverse generative models. AIGI-Holmes (Zhou et al., 2025) fuses NPR with semantic-rich pretrained CLIP-ViT embedding within MLLM to deliver both robust generalisation to unseen AI-generated images and human verifiable explanations.

However, our preliminary analysis (Sect. 3) finds that directly concatenating artifact features and semantic features as input to the LLM leads to suboptimal fusion, since the artifact features exhibit high intra-feature similarity, resulting in an almost uniform attention map that weakens interaction. To address this limitation, we introduce *TranX-Adapter* in Sect. 4, which promotes a more profound integration of artifact and semantic representations, thereby advancing the effectiveness and robustness of AIGI detection.

## 3. Pilot Study

In this section, we conduct a pilot study to demonstrate that self-attention fusion of artifact and semantic features within the LLM is suboptimal. Following AIGI-Holmes (Zhou et al., 2025), we employ NPR (Tan et al., 2024) as the artifact encoder and integrate it into LLaVA-1.6-mistral 7B (Liu et al., 2024), while CLIP-ViT provides the semantic features. The two types of features are concatenated and fed into the LLM for joint training. In Sect. 3.1, we analyze the intrinsic distributional differences between artifact and semantic features. In Sect. 3.2, attention map visualization reveals that when the LLM performs self-attention fusion, the Artifact → Semantic interaction exhibits `attention dilution` (Zhang et al., 2024), where the attention distribution becomes nearly uniform, indicating ineffective feature fusion. Finally, by quantifying the mean significance of information flow, we further confirm that existing self-attention fusion within the LLM remains suboptimal (Sect. 3.2).

### 3.1. Uniform Artifact and Variance Semantic

In AIGI detection, artifact features are essential for distinguishing synthetic images from authentic ones (Tan et al., 2024; Chen et al., 2025; Qian et al., 2020). The NPR model exemplifies this principle by leveraging local pixel dependencies to capture subtle inconsistencies that signal synthetic content. Specifically, NPR first upsamples the input image through bilinear interpolation, subsequently downsamples it, and computes the residual between the reconstructed and original images, which is then processed by ResNet (He et al., 2016).

As illustrated in Figure 2 (a), the NPR input image primarily captures structural cues such as edges and textures, leading to substantial redundancy among spatial patches. In contrast, MLLMs typically employ CLIP-ViT as the visual encoder, which directly processes the original image to extract rich and diverse semantic representations. This inherent discrepancy is further manifested at the feature level. As shown in Figure 3, we examine the distributional characteristics of artifact and semantic features by computing the L2 norm and cosine similarity image patches derived from each visual encoder. The results reveal that CLIP-ViT features exhibit significantly higher inter-patch variance, whereas NPR features demonstrate strong intra-feature similarity, indicating that *artifact representations are more homogeneous and less discriminative in feature space*.

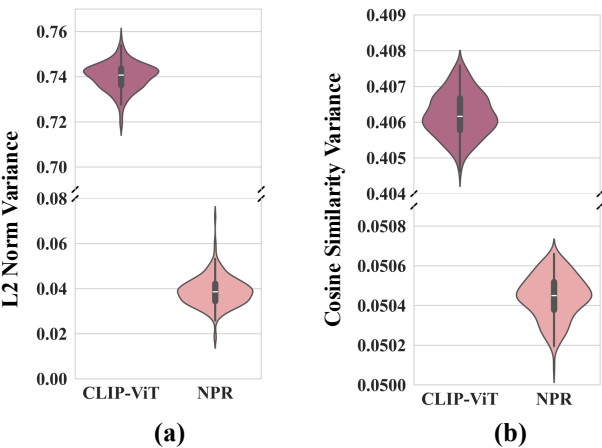

(a)             (b)

*Figure 3.* **Distributional comparison of representational variances between CLIP-ViT and NPR:** (a) variance of L2 norms and (b) variance of cosine similarities across image patches.

### 3.2. Semantic Limitations in Artifact Cue Extraction

We previously observed that the artifact features extracted by NPR exhibit strong intra-feature similarity. Consequently, during self-attention, this high similarity causes the atten-

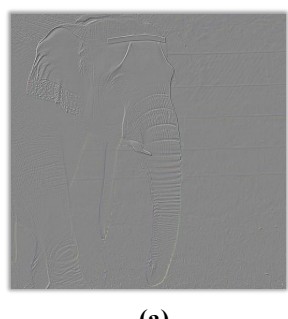 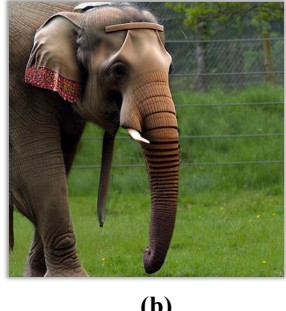

(a)                 (b)

*Figure 2.* **Comparison between the processed NPR input (a) and the CLIP-ViT input image**[1] **(b), where NPR highlights local pixel interdependencies for synthetic image detection.**

---

[1]The image is sourced from BiasFree (Guillaro et al., 2025)

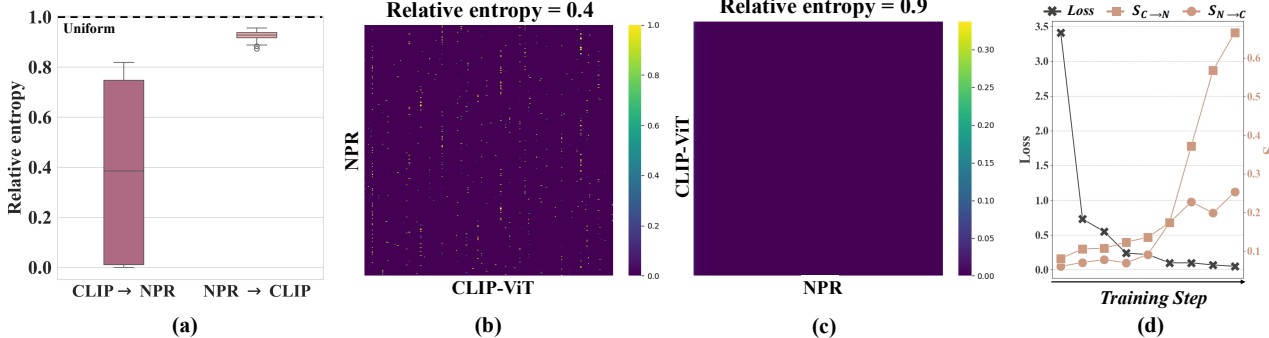

*Figure 4.* **Comparison of cross-encoder interactions between CLIP-ViT and NPR.** (a) Relative entropy of attention maps. Higher relative entropy values, approaching 1, indicate that the distribution is closer to uniform. (b) Attention map for the direction $CLIP \rightarrow NPR$, where the `query` originates from NPR features and the `key` and `value` are derived from CLIP-ViT features. (c) Attention map for the direction $NPR \rightarrow CLIP$, where the `query` corresponds to CLIP-ViT features and the `key` and `value` are obtained from NPR features. (d) Relationship between the training loss and the information flow metrics $S$.

tion map, when semantic features are used as *queries* and artifact features as *keys*, to collapses into an almost uniform distribution after softmax normalization. As illustrated in Figure 4 (a), the relative entropy of the CLIP-ViT → NPR is notably lower, indicating a more concentrated distribution as shown in Figure 4 (b). In contrast, the NPR → CLIP-ViT exhibits higher relative entropy, resulting in an almost uniform pattern as shown in Figure 4 (c) and leading to the phenomenon of `attention dilution`. However, the critical forgery cues embedded in artifact features typically reside in high-frequency regions (Zhong et al., 2023a), and this diluted attention impedes the effective transmission of critical forgery cues from artifact to semantic.

To further substantiate this observation, we follow Wang et al. (2023a) and introduce two quantitative metrics: (1) $S_{N \rightarrow C}$ denotes the mean significance of the information flow from NPR to CLIP-ViT, and (2) $S_{C \rightarrow N}$ denotes the mean significance of information flow from CLIP-ViT to NPR. The formal definition of information flow $S$ is provided in the Appendix A.1. As shown in Figure 4 (d), the information flow $S_{N \rightarrow C}$ is substantially lower than $S_{C \rightarrow N}$, indicating that ***the artifact information encoded by NPR is difficult to transfer into the semantic feature space***.

To confirm that this phenomenon is not specific to NPR, we further observe similar behavior in VAE-based artifact encoder (Cheng et al., 2025). Due to space constraints, the corresponding experimental results are presented in the Appendix B.1. These findings highlight that effectively bridging artifact and semantic features remains a critical challenge for MLLM performance in AIGI detection tasks.

## 4. Method

With the insights in Sect. 3, we introduce a lightweight fusion adapter, termed ***TranX-Adapter***, which facilitates

more effective fusion of semantic and artifact features. In this section, we introduce the ***TranX-Adapter*** in detail.

### 4.1. Preliminary

Given an image, we first extract artifact features $F_{art} \in \mathbb{R}^{N \times D}$ and semantic features $F_{sem} \in \mathbb{R}^{M \times D}$ from artifact encoder (*e.g.*, NPR) and semantic encoder (*e.g.*, CLIP-ViT) respectively. The $N$ and $M$ represent the lengths of the sequences of visual patches for artifact features $F_{art}$ and semantic features $F_{sem}$. The $D$ is the feature dimension of each visual patches. In fact, we employ separate Multi-Layer Perceptron (MLP) to align the feature dimensions of the artifact features and semantic features. And then the artifact features $F_{art}$ and semantic features $F_{sem}$ are fused by our ***TranX-Adapter***, which enables bidirectional fusion via *Task-Aware Optimal-Transport Fusion* (Artifact → Semantic) and *X-Fusion* (Semantic → Artifact). The fused visual features are projected to the LLM as visual tokens $\mathcal{V}$ and paired with a textual prompt $T$. To train our ***TranX-Adapter***, we apply the language modeling objective $\mathcal{L}$ to predictions of text tokens:

$$\mathcal{L} = -\sum_t \log P_\theta(y_t | \mathcal{V}, T, y_{<t}). \quad (1)$$

### 4.2. Task-Aware Optimal-Transport Fusion

In this part, we first provide the motivation of our *Task-Aware Optimal-Transport Fusion (TOP-Fusion)*, and then introduce it in detail.

**Motivation of TOP-Fusion.** The primary objective of this module is to transfer discriminative artifact cues into semantic features, compensating for forgery cues that the semantic space fails to capture. In Sect. 3, we observe that the Artifact → Semantic interaction suffers from

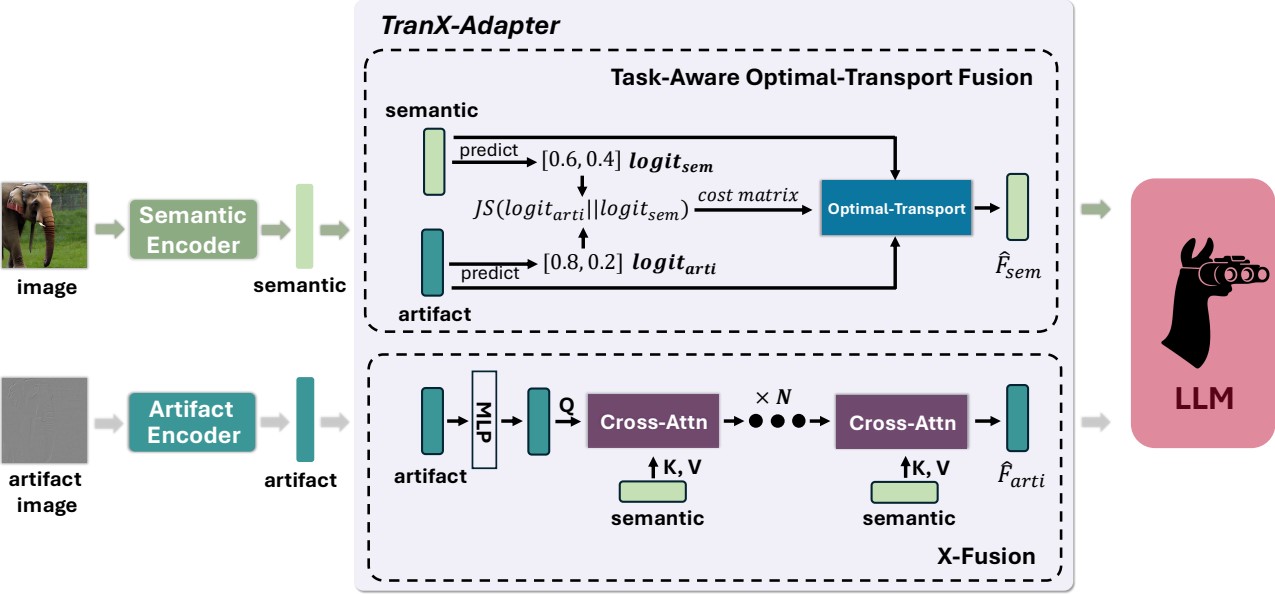

Figure 5. **Overview of the proposed *TranX-Adapter*.** Our *TranX-Adapter* consists of two complementary fusion modules. The *Task-Aware Optimal-Transport Fusion (TOP-Fusion)* aligns artifact and semantic feature predictions by computing JS divergence between their logits and transferring artifact features into the semantic features through optimal-transport, yielding an enhanced semantic feature $\hat{F}_{sem}$. The *X-Fusion* module transfers semantic features into artifact features via multi-layer cross-attention, producing $\hat{F}_{art}$. The fused representations are finally fed into the Large Language Model (LLM) for detection.

`attention dilution`, where the attention distribution becomes nearly uniform due to the high intra-feature similarity of artifact features. Prior studies (Zhong et al., 2023a; Frank et al., 2020) indicate that generative models predominantly introduce artifacts in the high-frequency regions of synthetic images, leading to varying artifact intensity across patches. To capture this, we replace the artifact features with the probability of each patch being predicted as fake, and similarly map semantic features into the same probability space. The fusion process is guided to emphasize regions exhibiting substantial discrepancies between the artifact and semantic features.

**TOP-Fusion.** The processes of our *TOP-Fusion* is illustrated in Figure 5. As discussed above, we first convert both artifact and semantic features into the probability of each patch being predicted as "fake". For artifact encoder, we use the prediction head from the artifact encoder training to obtain logits over the detect "fake". For the semantic encoder, when CLIP-ViT is used, we encode the text prompt "*A photo is a fake.*" with CLIP's text encoder (Devlin et al., 2019) and compute its similarity to the CLIP-ViT visual features. For semantic encoders without an aligned text encoder, we use a pretrained prediction head (see Appendix A.3) to obtain logits for fake detection. The complementary "real" logit is obtained as $1 - \sigma(\cdot)$. The $\sigma(\cdot)$ is the sigmoid function. We denote $P$ and $S$ as the probability distributions corresponding to the artifact and semantic features, respectively. Then,

we quantify the discrepancy between artifact and semantic using the Jensen-Shannon (JS) divergence, *i.e.*, $JS(S||P)$.

Next, we transfer the artifact features into the semantic space based on this discrepancy. Specifically, we employ Optimal-Transport (Gabriel & Marco, 2019; Santambrogio, 2015), using the $JS(S||P)$ as the cost matrix, and obtain the transport plan $\gamma$ via the Sinkhorn algorithm (Cuturi, 2013). Since the patches with substantial discrepancies should receive greater emphasis during feature transfer, we adopt the negative JS divergence, *i.e.*, $-JS(S||P)$. Finally, we use the transport plan $\gamma$ to transfer the artifact features into semantic features:

$$F_{art \to sem} = \gamma(F_{art}W_{art}), \qquad (2)$$

$$\hat{F}_{sem} = F_{sem} + \texttt{MLP}(F_{art \to sem}), \qquad (3)$$

where $W_{art} \in \mathbb{R}^{D \times d}$ are learned linear projections into the shared space, and $\texttt{MLP}(\cdot)$ is the Multi-Layer Perceptron that adapts the $F_{art \to sem}$ before fusion.

### 4.3. X-Fusion

In this part, we first provide the motivation of our *X-Fusion*, and then introduce it in detail.

**Motivation of X-Fusion.** Previously, we introduced *TOP-Fusion* to achieve the interaction from Artifact $\to$ Semantic. The most straightforward approach is to directly integrate *TOP-Fusion* into the LLM. However, we argue that such

a strategy is inefficient for two main reasons: (1) modifying the internal architecture of the LLM may disrupt its inherent knowledge, and (2) it requires training the LLM. In fact, our experiments (Sect. 5.7) reveal that the interactions among different visual features within the LLM primarily occur in the shallow layers. Therefore, we concentrate the interaction between artifact and semantic features within a lightweight adapter and fine-tune only this adapter during training, avoiding the need to update the LLM. Consequently, we employ a cross (X)-attention mechanism to achieve the Semantic $\to$ Artifact interaction.

**X-Fusion.** Given the artifact features $F_{art}$ and semantic features $F_{sem}$, both feature types are first projected into a shared latent space of dimension $d$ through linear transformations:

$$\tilde{F}_{art} = F_{art}\tilde{W}_{art}, \quad \tilde{F}_{sem} = F_{sem}\tilde{W}_{sem}, \quad (4)$$

where $\tilde{W}_{art}$, and $\tilde{W}_{sem} \in \mathbb{R}^{D \times d}$. Subsequently, a stack of cross-attention layers is employed to enable fine-grained semantic injection. In each layer $\ell$, the artifact features serve as the *query*, while semantic features act as *key* and *value*:

$$Q = \tilde{F}_{art}W_Q, \ K = \tilde{F}_{sem}W_K, \ V = \tilde{F}_{sem}W_V, \quad (5)$$

$$H = \texttt{softmax}(\frac{QK^\top}{\sqrt{d}})V, \quad (6)$$

$$X = \tilde{F}_{art} + H, \quad (7)$$

$$F_{sem \to art} = X + \texttt{MLP}(X), \quad (8)$$

where $W_Q, W_K$ and $W_V \in \mathbb{R}^{d \times d}$ are the parameter matrics. In this formulation, the artifact features actively retrieve complementary semantic cues from the semantic features, facilitating semantic-aware enhancement. After the final cross-attention block, the updated representation is normalized and mapped back to the original feature dimension:

$$\hat{F}_{art} = F_{art} + \texttt{MLP}(F_{sem \to art}). \quad (9)$$

This process yields a refined artifact features that effectively incorporates semantic guidance while preserving the original feature characteristics.

# 5. Experiments

In this section, we first present the experimental setup, including implementation details and evaluation protocols, followed by the experiments and analysis. Further experimental results, including the generalization analyses using different artifact encoders, and more ablation studies are provided in Appendix B. Case studies are provided in the Appendix C.

## 5.1. Experimental Setup

**Implementation Details.** Following (Zhou et al., 2025), we adopt NPR (Tan et al., 2024) as the artifact encoder to extract artifact features, while using the visual encoder of the MLLM as the semantic encoder to capture high-level visual semantics. However, our *TranX-Adapter* also generalizes to other artifact encoders (see Appendix B.1). We use LLaVA-1.6-mistral 7B and Qwen3-VL (2B and 4B) as the base models. Unless otherwise specified, the parameter of MLLMs are kept frozen during training. Further implementation details are provided in Appendix A.

**Evaluation Protocols and Dataset.** We adopt two widely recognized evaluation protocols. In **Protocol I**, we follow the GenImage (Zhu et al., 2023b) benchmark and train the model on real images (*e.g.*, ImageNet (Deng et al., 2009)) and SD v1.4 (Rombach et al., 2022) generate images, and evaluate the trained model across eight different generators (Brock et al., 2018; Nichol et al., 2021; Gu et al., 2022; Dhariwal & Nichol, 2021; Midjourney, 2022; Wukong, 2022; Rombach et al., 2022). In **Protocol II**, the model is trained on data from diverse generative models and evaluated on comprehensive benchmarks containing challenging samples from modern generators. This protocol uses the Chameleon (Yan et al., 2025) and RRDataset (Li et al., 2025).

## 5.2. Comparison with AIGI Detection Methods

**Comparison on GenImage (Protocol I).** As shown in Table 1, *TranX-Adapter* consistently boosts performance across all evaluated models, including LLaVA-1.6-mistral 7B, Qwen3-VL 2B, and Qwen3-VL 4B, demonstrating strong model-agnostic effectiveness. Additionally, many existing methods overfit to specific generators and suffer significant performance drops on unseen ones, whereas our method maintains consistently high accuracy across generators, demonstrating strong robustness and transferability.

**Comparison on Chameleon (Protocol II).** To mitigate potential biases arising from training configurations, such as generator diversity and image category imbalance, we evaluate our method alongside existing detectors under varied training conditions. As shown in Table 2, prior approaches exhibit limited transferability, with accuracies ranging from 55% to 72%. In contrast, LLaVA-1.6-mistral 7B w/ our *TranX-Adapter* achieves 75.8% accuracy when trained solely on SD v1.4 and 85.1% when trained on the full GenImage dataset, significantly outperforming all baselines. These results underscore the strong generalization and robustness of the proposed *TranX-Adapter*.

**Comparison on RRDataset (Protocol II).** Compared with Chameleon, RRDataset (Li et al., 2025) encompasses seven distinct scenes and four re-digitization processes, offering a more comprehensive assessment of the generalization capability of AIGI detectors. As presented in Table 3, we evaluate both zero-shot MLLMs and conventional detectors

*Table 1.* **Cross-model accuracy performance on the GenImage Dataset.** Accuracy (%) of different detectors (rows) in distinguishing real images from those produced by various generative models (columns). The best result are marked in **bold**. "†" indicates direct concatenation of artifact and semantic features following Zhou et al. (2025).

| Method | Midjourney | SD v1.4 | SD v1.5 | ADM | GLIDE | Wukong | VQDM | BigGAN | Mean |
|---|---|---|---|---|---|---|---|---|---|
| CNNSpot (Wang et al., 2020) | 52.8 | 96.3 | 95.9 | 50.1 | 39.8 | 78.6 | 53.4 | 46.8 | 64.2 |
| F3Net (Qian et al., 2020) | 50.1 | **99.9** | **99.9** | 49.9 | 50.0 | **99.9** | 49.9 | 49.9 | 68.7 |
| DIRE (Wang et al., 2023b) | 60.2 | 99.9 | 99.8 | 50.9 | 55.0 | 99.2 | 50.1 | 50.2 | 70.7 |
| GenDet (Zhu et al., 2023a) | 89.6 | 96.1 | 96.1 | 58.0 | 78.4 | 92.8 | 66.5 | 75.0 | 81.6 |
| PatchCraft (Zhong et al., 2023b) | 79.0 | 89.5 | 89.3 | 77.3 | 78.4 | 89.3 | 83.7 | 72.4 | 82.3 |
| UnivFD (Ojha et al., 2023) | 73.2 | 84.2 | 84.0 | 55.2 | 76.9 | 75.6 | 56.9 | 80.3 | 73.3 |
| AIDE (Yan et al., 2025) | 79.4 | 99.7 | 99.8 | 78.5 | **91.8** | 98.7 | 80.3 | 66.9 | 86.9 |
| NPR (Tan et al., 2024) | 89.8 | 90.7 | 90.7 | 84.6 | 90.3 | 90.7 | 87.0 | 81.8 | 88.3 |
| LLaVA-1.6-mistral 7B† (Liu et al., 2024) | 88.6 | 94.0 | 94.0 | 80.3 | 86.8 | 93.4 | 82.6 | 76.2 | 87.3 |
| AIGI-Holmes (Zhou et al., 2025) | 81.6 | 91.3 | 91.4 | **88.4** | 91.5 | 89.5 | **90.9** | **94.5** | 89.8 |
| Qwen3-VL 2B† (Bai et al., 2025) | 88.1 | 87.3 | 84.1 | 78.6 | 81.7 | 81.7 | 84.1 | 72.2 | 82.2 |
| Qwen3-VL 4B† (Bai et al., 2025) | 87.3 | 98.4 | 96.0 | 65.1 | 91.3 | 96.0 | 73.8 | 78.6 | 85.8 |
| *w/ our TranX-Adapter* | | | | | | | | | |
| Qwen3-VL 2B | 90.5 | 97.6 | 96.0 | 83.3 | 89.7 | 92.9 | 82.5 | 71.4 | 88.0 |
| Qwen3-VL 4B | 92.1 | 97.6 | 97.6 | 81.0 | 83.3 | 94.4 | 84.9 | 87.3 | 89.8 |
| **LLaVA-1.6-mistral 7B** | **94.9** | 96.4 | 96.4 | 87.0 | 88.0 | 94.9 | 90.1 | 85.9 | **91.9** |

*Table 2.* **Cross-dataset accuracy performance on the Chameleon testset.** "†" denotes the setting following Zhou et al. (2025), where the artifact features and semantic features are directly concatenated.

| Method | Training Set | |
|---|---|---|
| | SDv1.4 | All GenImage |
| UnivFD (Ojha et al., 2023) | 55.6 | 60.4 |
| DIRE (Wang et al., 2023b) | 59.7 | 57.8 |
| PatchCraft (Zhong et al., 2023b) | 56.3 | 55.7 |
| NPR (Tan et al., 2024) | 58.1 | 57.8 |
| LLaVA-1.6-mistral 7B† (Liu et al., 2024) | 69.4 | 81.9 |
| AIDE (Yan et al., 2025) | 62.6 | 65.8 |
| PatchAll/CLIP (Yang et al., 2025) | 63.9 | 69.3 |
| PatchAll/DINOv2 (Yang et al., 2025) | 66.6 | 72.1 |
| Qwen3-VL 2B† (Bai et al., 2025) | 68.5 | 78.8 |
| Qwen3-VL 4B† (Bai et al., 2025) | 69.4 | 78.3 |
| *w/ our TranX-Adapter* | | |
| Qwen3-VL 2B | 71.8 | 82.3 |
| Qwen3-VL 4B | 72.6 | 83.6 |
| **LLaVA-1.6-mistral 7B** | **75.8** | **85.1** |

*Table 3.* **Cross-dataset accuracy performance on the RRDataset.** The "*Ori.*" represents the Original, "*Trans.*" represents the Transmission, and "*Re.*" represents the Re-digitization. "†" indicates the direct concatenation of artifact and semantic features (Zhou et al., 2025).

| Method | RRDataset | | | |
|---|---|---|---|---|
| | Ori. | Trans. | Re. | Avg. |
| *MLLMs (Zero-shot)* | | | | |
| GPT-4o (Achiam et al., 2023) | 94.5 | 84.7 | 73.1 | 84.1 |
| Claude-3.7-sonnet (Ant) | 89.9 | 83.8 | 73.9 | 82.5 |
| Gemini-2-flash (Team et al., 2023) | 85.3 | 74.8 | 71.8 | 77.3 |
| Qwen2VL-72B (Wang et al., 2024) | 59.9 | 56.4 | 59.8 | 58.7 |
| *Detectors (Fine-tuned)* | | | | |
| DIRE (Wang et al., 2023b) | 94.0 | 94.1 | 50.2 | 79.4 |
| AIDE (Yan et al., 2025) | 79.0 | 76.8 | 79.6 | 78.4 |
| GramNet (Liu et al., 2020) | 78.0 | 77.6 | 70.8 | 75.4 |
| CNNSpot (Wang et al., 2020) | 80.8 | 77.3 | 64.9 | 74.3 |
| NPR (Tan et al., 2024) | 72.7 | 62.6 | 65.6 | 67.0 |
| C2P-CLIP (Tan et al., 2025) | 57.4 | 64.2 | 54.2 | 58.6 |
| LLaVA-1.6-mistral 7B† (Liu et al., 2024) | 94.8 | 69.3 | 85.5 | 83.2 |
| Qwen3-VL 2B† (Bai et al., 2025) | 88.9 | 89.9 | 68.8 | 82.5 |
| Qwen3-VL 4B† (Bai et al., 2025) | 96.0 | 89.5 | 71.3 | 85.6 |
| *w/ our TranX-Adapter* | | | | |
| Qwen3-VL 2B | 97.5 | 95.3 | 78.3 | 88.9 |
| **Qwen3-VL 4B** | **98.1** | **95.5** | **79.0** | **90.9** |
| LLaVA-1.6-mistral 7B | 96.6 | 93.0 | 77.1 | 88.9 |

on RRDataset. Our ***TranX-Adapter*** integrated with Qwen3-VL 4B achieves the highest accuracy of 90.9%, surpassing the strongest baseline (Qwen3-VL 4B, 85.6%) by 5.3% and GPT-4o by +6.8%. These results demonstrate the superior generalization of our approach for AIGI detection.

### 5.3. Ablation Study

To elucidate the contribution of each component within our ***TranX-Adapter***, we perform ablation studies on GenImage using LLaVA-1.6-mistral 7B. As presented in Table 4, incorporating the artifact encoder to provide auxiliary forgery cues yields a 4.6% improvement in accuracy. Introducing *X-Fusion* and *TOP-Fusion* individually brings further gains of 2.3% and 3.2%, respectively. When both fusion strategies are jointly applied, the model attains an accuracy of 91.9%,

confirming that the two fusion mechanisms complement each other and jointly enhance detection performance.

**Fusion Strategy Ablation.** To further examine how fusion strategies affect different information transfer processes, we consider three variants: 1) applying *TOP-Fusion* to both transfer processes, 2) applying *X-Fusion* to both transfer processes, and 3) our ***TranX-Adapter***. Experiments are conducted with Qwen3-VL 2B, trained on GenImage and evaluated on Chameleon. As shown in Table 5, uniformly usiing either *TOP-Fusion* or *X-Fusion* for both processes yields inferior performance compared with ***TranX-***

*Table 4.* **Ablation study of different module in *TranX-Adapter*.** The first row shows the results of LLaVA-1.6-mistral 7B. We integrate NPR, apply *X-Fusion* and *TOP-Fusion* separately, and then use the full bidirectional fusion strategy.

| Artifact Encoder | X-Fusion | TOP-Fusion | Accuracy |
|:---:|:---:|:---:|:---:|
|  |  |  | 82.3 |
| ✓ |  |  | 86.0 |
| ✓ | ✓ |  | 89.3 |
| ✓ |  | ✓ | 90.3 |
| ✓ | ✓ | ✓ | **91.9** |

*Adapter*. This result demonstrates that the two transfer processes require distinct fusion mechanisms, validating the asymmetric design of *TranX-Adapter*.

*Table 5.* **Ablation study on artifact-semantic fusion strategies.** We compare different fusion strategy for Semantic → Artifact and Artifact → Semantic information transfer. Our asymmetric design, using X-Fusion for Semantic → Artifact and TOP-Fusion for Artifact → Semantic, achieves the best accuracy.

| Semantic→Artifact | Artifact→Semantic | Accuracy |
|:---:|:---:|:---:|
| TOP-Fusion | TOP-Fusion | 78.9 |
| X-Fusion | X-Fusion | 81.3 |
| X-Fusion | TOP-Fusion | **82.3** |

## 5.4. Robustness Studies

We conduct comprehensive robustness evaluations to assess the reliability of our approach under common image corruptions. Specifically, we compare our *TranX-Adapter* with three representative detectors, namely NPR (Tan et al., 2024), CO-SPY (Cheng et al., 2025), and Effort (Yan et al.), across JPEG compression (quality factors $Q = 95, 90, 75, 50$), Gaussian blur (standard deviations $\sigma = 1.0, 2.0, 3.0, 4.0$), and image resizing (scales $S = 0.5, 1.0, 1.5, 2.0$) on the SDv1.4 subset of the GenImage. Additionally, we also conduct experiment on LLaVA-1.6-mistral-7B (Liu et al., 2024). As shown in Figure 6, our method demonstrates strong robustness across all corruption types, preserving high accuracy even under extreme compression, severe blur, and substantial scale variation. Notably, our *TranX-Adapter* maintains approximately 90% accuracy in the most challenging scenarios.

## 5.5. Effect of Different Cost Matrices.

In *TOP-Fusion*, we employ the Jensen-Shannon (JS) divergence to quantify the discrepancy between the artifact and semantic probability distributions. To further investigate the effect of different cost matrix designs, we replace the JS divergence with *cosine similarity (COS)* and *Kullback-Leibler (KL)* divergence, respectively. For cosine similarity, we directly compute the cosine similarity between artifact and semantic features as the cost matrix. For KL divergence,

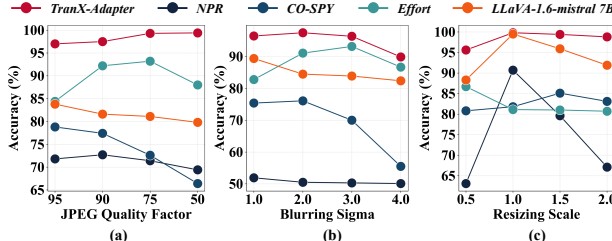

*Figure 6.* **Robustness analysis on JPEG compression, Resizing and Gaussian blur.** (a) Impact of JPEG compression, (b) Impact of Gaussian blur and (c) Impact of Resizing.

we use the artifact and semantic logits as inputs to compute the divergence. We train the models on GenImage SD v1.4 and evaluate them on Chameleon and GenImage.

As shown in Table 6, using cosine similarity between artifact and semantic features as the cost matrix in *TOP-Fusion* leads to substantially lower accuracy on both Chameleon and GenImage, compared to using discrepancies between their task-specific prediction probabilities. This is because the high intra-feature similarity of artifact features induces a near-uniform cosine similarity pattern, limiting the transfer of fine-grained artifact cues. In contrast, discrepancy measures computed in the prediction space exhibit stronger patch-wise variations (see Sect. 5.7), enabling more effective fusion. Moreover, compared to KL divergence, JS divergence provides more stable training and consistently achieves better performance.

*Table 6.* **Ablation study of different cost matrices in *TranX-Adapter*.** The "*COS*" denotes the cosine similarity. The "*KL*" denotes the Kullback-Leibler divergence and "*JS*" denotes the Jensen-Shannon divergence. The "Δ ↑" represents the performance gains of our *TranX-Adapter* against the baseline.

| | Chameleon ↑ | GenImage ↑ | Δ ↑ |
|:---|:---:|:---:|:---:|
| Qwen3-VL 2B | 68.5 | 82.2 | 0.0 |
| *TOP-Fusion w/ COS* | 69.8 | 83.7 | +1.4 |
| *TOP-Fusion w/ KL* | 70.8 | 86.8 | +3.5 |
| ***TOP-Fusion w/ JS*** | ***71.8*** | ***88.0*** | **+4.6** |

## 5.6. Comparison with PEFT Methods

To evaluate training efficiency, we compare our *TranX-Adapter* with Parameter-Efficient Fine-Tuning (PEFT) methods, including LoRA (Hu et al.) and Adapter (Houlsby et al., 2019), on the Chameleon dataset, using GenImage-SD v1.4 as the training set. As shown in Table 7, our *TranX-Adapter* achieves performance comparable to full fine-tuning while using only a tiny-scale of the parameters.

## 5.7. Why does Our TranX-Adapter Work?

In Sect.3, we observe that the artifact features exhibit high intra-feature similarity, preventing effective injection of artifact information into the semantic features. In this section,

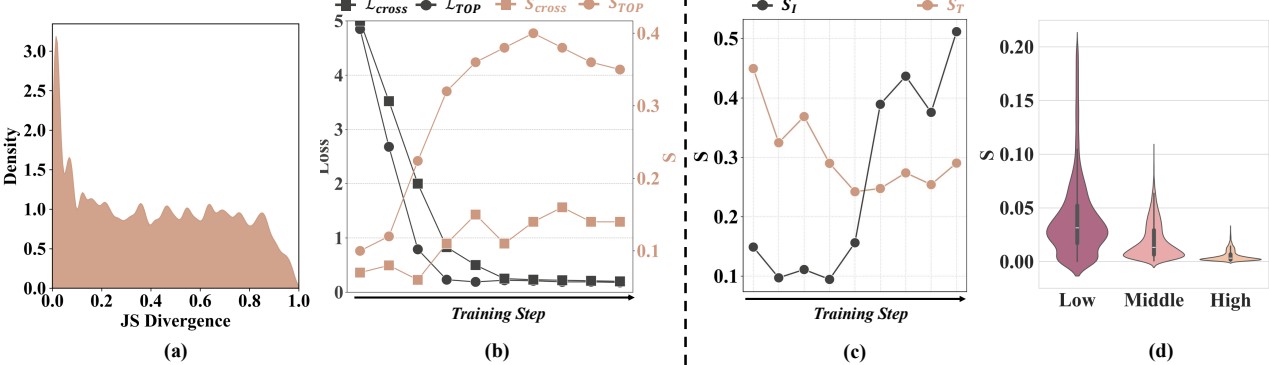

*Figure 7.* **Overview of JS divergence and the evolution of information-flow significance across training.** (a) Distribution of JS divergence. (b) Illustrates the relationship between the training loss and the information-flow significance $S$. (c) Comparison of image and text information-flow significance and (d) layer-wise distribution of $S$ across the LLM, where "Low" indicates layers near the input.

*Table 7.* **Comparison with PEFT methods on Chameleon dataset**, where "Params." denotes the number of learnable parameters measured in millions.

| Method | Params. (M) | Acc. |
|---|---|---|
| Full | 7261 | **76.8** |
| LoRA (Hu et al.) | 40 | 69.1 |
| | 160 | 74.4 |
| Adapter (Houlsby et al., 2019) | 40 | 69.4 |
| | 160 | 72.5 |
| *TranX-Adapter* | 40 | 73.8 |
| | 160 | 75.8 |

we analyze why *TranX-Adapter* is effective from the following two perspectives:

*1) Why can TOP-Fusion transfer artifact information into semantic features more effectively?* We design *TOP-Fusion* to selectively amplify regions where artifact and semantic diverge in AIGI detection. To achieve this, we compute the JS divergence between their prediction probabilities and adopt it as the cost matrix within the optimal transport. As illustrated in Figure 7 (a), the JS divergence exhibits low JS divergence regions dominate with high density, whereas high JS divergence regions are relatively sparse. This non-uniformity easily guides the model to focus on patches with substantial discrepancies. To further assess the advantages of *TOP-Fusion*, we compare it against a standard cross-attention by substituting *TOP-Fusion* with cross-attention. We denote the mean significance of information flow (Wang et al., 2023a) from artifact to semantic features as $S_{TOP}$ and $S_{cross}$ for *TOP-Fusion* and cross-attention, respectively. As shown in Figure 7 (b), $S_{TOP}$ consistently surpasses $S_{cross}$, and *TOP-Fusion* achieves a lower training loss under identical training steps. These findings demonstrate that *TOP-Fusion more effectively highlights regions with substantial discrepancies between artifact and semantic features, thereby enabling more efficient fusion.*

*2) Why can TranX-Adapter remain effective with only a small number of trainable parameters?* To answer this question, it is essential to clarify what the LLM primarily learns during training. We therefore examine how different features, including text embeddings, artifact features, and semantic features, interact within the LLM. A naive concatenation strategy is adopted, where all three feature types are provided jointly as input while fine-tuning the LLM. We denote $S_T$ and $S_I$ as the mean significance of information flow (Wang et al., 2023a) from text embeddings and visual features to the final output, respectively. As shown in Figure 7 (c), the model increasingly depends on visual features as training progresses, indicating that *the LLM places greater emphasis on visual feature interactions.* Furthermore, Figure 7 (d) shows that *these visual feature interactions predominantly emerge in the shallow layers (corresponding to "Low") of the LLM*. This behavior aligns with the design of *TranX-Adapter*, which promotes effective visual feature interaction via *TOP-Fusion* and *X-Fusion* with minimal additional parameters.

## 6. Conclusion

In this paper, we introduce *TranX-Adapter*, a lightweight fusion adapter that effectively integrates semantic and artifact features to strengthen the AIGI detection capability of MLLMs. Extensive experiments across diverse benchmarks demonstrate that *TranX-Adapter* delivers robust performance and superior generalization. Our analysis reveals three key insights: (1) high intra-feature similarity of artifact features leads to `attention dilution` under self-attention, while discrepancy-aware fusion is more effective; (2) the LLM increasingly relies on visual information during training and (3) artifact-semantic fusion predominantly occurs in shallow layers. Overall, *TranX-Adapter* improves artifact utilization in MLLMs, paving the way for future work on AIGI localization and explainability.

## Acknowledgements

This work is supported by the National Natural Science Foundation of China (Grant No. U23A20318 and 62276195), the New Cornerstone Science Foundation through the XPLORER PRIZE, the Foundation for Innovative Research Groups of Hubei Province (Grant No. 2024AFA017) and WHU-Kingsoft Joint Lab. The numerical calculations in this paper have been done on the supercomputing system in the Supercomputing Center of Wuhan University. This project is supported by the China Scholarship Council.

## Impact Statement

This work aims to safeguard public information integrity by enhancing the robustness of AI-generated image detection, directly addressing the growing threat of misinformation. Our proposed *TranX-Adapter* achieves superior generalization across diverse generators while utilizing a lightweight architecture, which significantly improves training efficiency and reduces computational energy costs compared to full MLLM fine-tuning. While we focus on defense, we acknowledge the potential risk of these techniques being used adversarially to evade detection and emphasize that such tools should support, rather than replace, human verification to mitigate the risks of classification errors.

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

# Appendix

This appendix presents a detailed implementation description of the proposed ***TranX-Adapter***, along with additional results from **generalization analyses**, **robustness studies**, **effect of different cost matrices** and **case studies**. The structure of the appendix is summarized as follows.

➤ Appendix A provides the implementation details of ***TranX-Adapter***, including the definition of information flow $S$, details of constructing training dataset and training hyperparameter settings. Specifically, we introduce the definition of information flow $S$ in Appendix A.1. Appendix A.2 presents details of constructing training dataset and details of the implementation of our ***TranX-Adapter*** in Appendix A.3. Finally, Appendix A.4 provides the details of the training hyperparameter settings.

➤ Appendix B provides additional experimental results, *e.g.*, the generalization analyses with VAE-based artifact encoder (Appendix B.1).

➤ Appendix C provides a qualitative analyses of the proposded ***TranX-Adapter***.

## A. Implementation Details

### A.1. Definition of Information Flow $S$

In Sect.3.2 and Sect. 5.7 of our paper, we employ the saliency technique (Simonyan et al., 2013) to reveal the intrinsic attention-interaction patterns between different feature types within the LLM. Following standard practice, we apply the Taylor expansion (Michel et al., 2019; Wang et al., 2023a) to compute the saliency score for each element in the attention matrix:

$$I_l = \sum_h |A_{h,l}^\top \frac{\partial \mathcal{L}(x)}{\partial A_{h,l}}|. \tag{10}$$

We denote $A_{h,l}$ as the attention matrix produced by the $h$-th head in the $l$-th layer, with $x$ representing the model input and $\mathcal{L}(x)$ the task-specific loss function (*e.g.*, the autoregressive loss used during LLM training). By averaging the saliency results from all attention heads, we derive a layer-wise saliency map $I_l$. Each element $I_l(i,j)$ quantifies how strongly information from the $j$-th token influences the $i$-th token. Inspecting $I_l$ reveals how features from distinct positions interact internally within the LLM. To systematically characterize these interactions, we further introduce quantitative indicator described below.

$S_{s \to t}$ denotes the mean significance of the information flow from $s$ to $t$:

$$S_{s \to t} = \frac{\sum_{(i,j) \in P_{s \to t}} I_l(i,j)}{|P_{s \to t}|}, \tag{11}$$

$$P_{s \to t} = \{(i,j) : i \in [1, P_s], j \in [1, P_t]\}, \tag{12}$$

where $P_s$ and $P_t$ denote the sequence lengths of the two different types of features. For example, $S_{N \to C}$ denotes the mean significance of the information flow from NPR to CLIP-ViT, where $P_s$ and $P_t$ correspond to the sequence lengths of the NPR and CLIP-ViT features, respectively.

### A.2. Constructing the Training Dataset.

To construct the training set, we employ ChatGPT (Achiam et al., 2023) to provide question templates (*e.g.*, "*Is there any indication that this photo is real or fake? Just answer Real or Fake.*") and restrict the model to output only "Real" or "Fake" during training. This constraint encourages the model to concentrate strictly on task-relevant signals, avoiding unnecessary computation on irrelevant tokens and substantially improving inference efficiency.

### A.3. Model Implementation Details

In our ***TranX-Adapter***, both artifact and semantic features are projected into a shared 512-dimensional latent space, *i.e.*, $d = 512$. For *TOP-Fusion*, we adopt the Sinkhorn algorithm (Cuturi, 2013), which iteratively updates the dual scalings using stable log-sum-exp operations over the KL-based cost kernel. For semantic encoders lacking an aligned text encoder, we introduce a lightweight pretraining strategy that attaches a linear prediction head and fine-tunes only this head. This design efficiently aligns the prediction head with the semantic encoder, enabling reliable logits that indicate whether an input image is "fake". For *X-Fusion*, we employ $N$ cross-attention layers, with $N = 3$ as the default configuration.

## A.4. Training Details

During our experiments, we observe that the model exhibited a tendency to overfit to image resolution (Chen et al., 2025). To address this issue, we resize all images fed into CLIP-ViT to $256 \times 256$. This adjustment effectively mitigates resolution-induced bias and simutaneously reduces GPU memory consumption during training. The detailed training configuration is provided in Table 8. For GenImage, we find that detection performance tends to overfit to image resolution. Therefore, during training, we apply data augmentation to a subset of images by cropping them to a fixed solution of $512 \times 512$ (Guillaro et al., 2025). For RRDataset, we follow Li et al. (2025) by pretraining on GenImage and fine-tune on the RRDataset-subset where images are resized to $512 \times 512$.

*Table 8.* **Training settings across different training sets.**

|  | *GenImage SD1.4* | *GenImage All* | *RRDataset* |
|---|---|---|---|
| Batch size | 24 | 24 | 24 |
| Learning rate | 5e-5 | 5e-5 | 5e-5 |
| Epoch | 3 | 1 | 9 |
| Warmup ratio | 0.03 | 0.03 | 0.03 |
| Optimizer | AdamW | AdamW | AdamW |

# B. More Experimental Results

## B.1. Generalization Analyses with Artifact Encoder

In Sect. 3 of our paper, our pilot study reveals that the artifact features extracted from NPR exhibit high intra-feature similarity, which limits the model's ability to effectively convey texture-level artifact cues into semantic features. To further validate this finding and demonstrate that the proposed ***TranX-Adapter*** is not restricted to NPR, we additionally conduct experiments using a VAE-based artifact encoder (Cheng et al., 2025; Kingma & Welling, 2013). Specifically, we follow Cheng et al. (2025) to capture the artifact cues, by feeding an image to the VAE and differencing the reconstructed image with the original one. And then we use ResNet-50 (He et al., 2016) to extract artifact features.

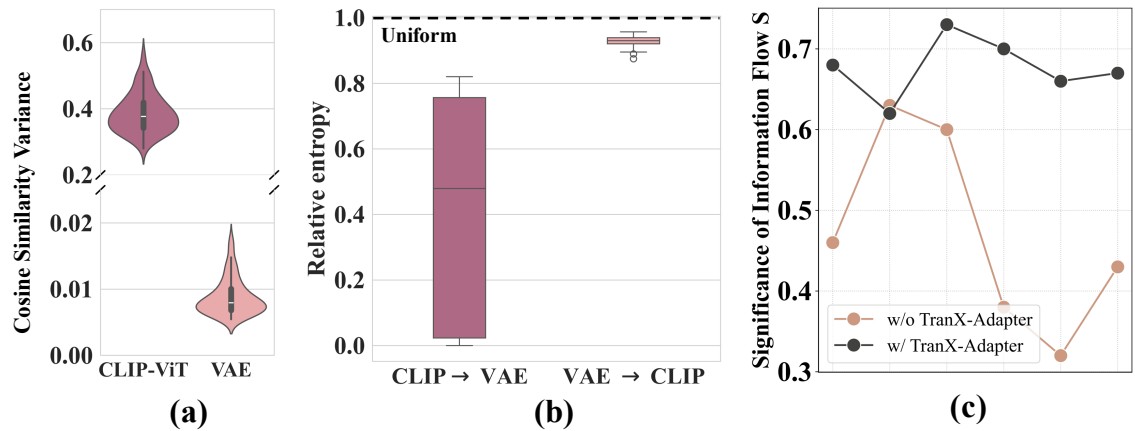

*Figure 8.* **Representational disparity between CLIP-ViT and VAE-based artifact encoder, and the corresponding information flow** $S$ **from Artifact → Semantic.** (a) variance of cosine similarities across image patches, (b) Relative entropy of attention maps and (c) visualization the significance of information flow $S$ from Artifact → Semantic.

As shown in Figure 8 (a), the artifact features extracted by the VAE-based encoder also exhibit pronounced intra-feature similarity. Furthermore, Figure 8 (b) shows that the VAE → CLIP yields consistently high entropy. These observations further corroborate the generality of our findings. As illustrated in Figure 8 (c), we visualize the mean significance of information flow $S_{art \rightarrow sem}$, and observe that the use of our ***TranX-Adapter*** substantially enhances the fusion between the artifact features and semantic features.

To verify that our ***TranX-Adapter*** is also applicable to the VAE-based artifact encoder, we conduct experiments on both

*Table 9.* **Cross-dataset accuracy under the Protocol II setting.** For Chameleon, we train on full GenImage dataset. The "*VAE*" denotes the VAE-based artifact encoder.

| | *Chameleon* | *RRDataset* | | | |
|---|---|---|---|---|---|
| | *(All GenImage)* | *Ori.* | *Trans.* | *Re.* | *Overall* |
| *LLaVA-1.6-mistral 7B +VAE* | | | | | |
| *w/o TranX-Adapter* | 79.3 | 96.5 | **94.8** | 76.8 | 89.4 |
| *w/ TranX-Adapter* | **83.8** | 96.5 | 94.4 | **82.7** | **91.2** |
| *Qwen3-VL 2B + VAE* | | | | | |
| *w/o TranX-Adapter* | 83.1 | 97.8 | 93.5 | 75.0 | 88.8 |
| *w/ TranX-Adapter* | **86.5** | **98.0** | **96.8** | **81.3** | **92.0** |

Chameleon (Yan et al., 2025) and RRDataset (Li et al., 2025) using LLaVA-1.6-mistral 7B and Qwen3-VL 2B. As shown in Table 9, our ***TranX-Adapter*** delivers consistent improvements across different artifact encoders.

## C. Case Study

Figure 9 visualizes examples on the Chameleon dataset using LLaVA-1.6-mistral 7B with NPR as the artifact encoder. Without ***TranX-Adapter***, attention is diffuse and fails to highlight meaningful artifact cues, leading to incorrect predictions. With ***TranX-Adapter***, attention becomes more structured and discriminative across both feature types, enabling correct identification of real and fake images.

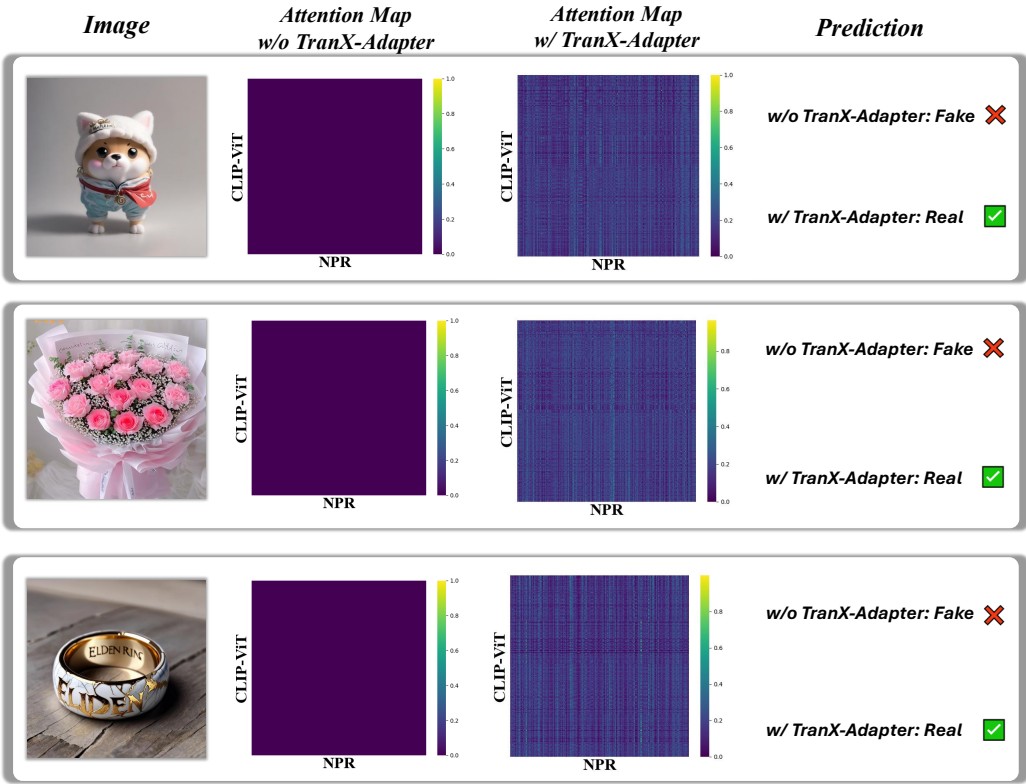

*Figure 9.* **Qualitative evidence illustrating the effect of *TranX-Adapter* on attention patterns and prediction quality.** For each input image, we visualize the attention maps derived from CLIP-ViT and NPR under the w/o ***TranX-Adapter*** and w/ ***TranX-Adapter***.

