# OpenReview forum: "TranX-Adapter: Bridging Artifacts and Semantics within MLLMs for Robust AI-generated Image Detection"
_ICML.cc/2026/Conference — ICML 2026 regular_

### Official Review · Reviewer_Haa2 · 2026-03-07

**Soundness:** 3
**Presentation:** 3
**Significance:** 3
**Originality:** 4
**Overall Recommendation:** 4
**Confidence:** 4

**Summary:**

In forgery detection tasks, simply concatenating artifact features (e.g., from the NPR encoder) with semantic features before feeding them into an LLM yields suboptimal results. This is because NPR's artifact features exhibit extremely high intra-feature similarity, leading to an almost uniform attention map after the Softmax operation, which prevents effective fusion of the two types of features and masks critical forgery cues. The authors propose TranX-Adapter, an asymmetric feature fusion architecture that eschews traditional dot-product attention in favor of calculating the JS divergence between artifact and semantic prediction probabilities as a cost matrix. It utilizes Optimal Transport to transfer artifact information into semantic features, achieving superior performance.

**Compliance With Llm Reviewing Policy:**

Affirmed.

**Final Justification:**

The author's reply resolved my concerns.

**Key Questions For Authors:**

1. CLIP is trained on a general domain via natural language supervision, and there is no evidence suggesting that CLIP can fully understand forgery. In TOP-Fusion, using CLIP's text tower to estimate the probability distribution for semantic features might lead to unreliable results. I hope the authors can provide further discussion on this.
2. In TOP-Fusion, a global calculation method is used when computing the JS divergence between semantic and artifact features. Consequently, the subsequent feature transfer could potentially occur between any two semantic and artifact features. Is this reasonable? A difference in the forgery probability distributions between artifact features at spatial location A and semantic features at spatial location B might simply be because forgery cues exist at A but not at B, rather than a conflict between the two. Is it reasonable to globally transfer artifact information to semantic features without considering the inherent locality of the semantic and artifact feature sequences? I hope the authors can provide further discussion on this.
3. Despite the absence of attention dilution issues, could using the TOP strategy in the Semantic $\rightarrow$ Artifact information transfer process within X-Fusion yield even better performance?

**Limitations:**

Yes

**Strengths And Weaknesses:**

Strengths
The paper is clearly written with extensive experiments. The proposed TranX-Adapter effectively addresses the problem of attention dilution.

Weaknesses
1. For MLLMs, minor changes in prompts often significantly impact prediction results. The paper uses templates generated by ChatGPT to constrain the MLLM to output "Real" or "Fake," yet lacks a systematic analysis of Prompt Robustness.
2. The paper performs excellently regarding JPEG compression, blurring, and resizing. Images in the real world may undergo multiple redistributions, screen re-digitization, or extreme low-light processing. These more complex physical-world noises could pose significant challenges to the texture-residual-based Artifact Encoder.
3. Some other details regarding the methodology design remain to be clarified (see the Key Questions For Authors).

---

> ### Author Rebuttal · Authors · 2026-03-31
>
> > Q1: Lacking a systematic analysis of prompt robustness.
>
> **R:** To evaluate the robustness of our method to prompt design, we consider two representative prompting strategies with different output constraints: **(1) a minimal format requiring the model to output only "Real" or "Fake"**, and **(2) a natural language format requiring a complete sentence**. We conduct controlled experiments using Qwen3-VL 2B on both GenImage and Chameleon. As shown below, performance is consistent across prompt variants, indicating robustness to prompt formulation. **The gains stem from the feature interaction mechanism rather than prompt engineering**.
>
> |                Prompt ID                                           | GenImage $\uparrow$ | Chameleon $\uparrow$ |
>    | - | - | - |
>    | (1)    | **88.0**     | **71.8**      |
>    | (2)| 87.5     | 70.6      |
>
> > Q2: Concerns about the robustness of the Artifact Encoder under complex real-world degradations beyond standard perturbations.
>
> **R:** In fact, the RRDataset already encompasses complex real-world scenarios, including **cross-patform redistribution**, **re-digitization** and **other complex environment**. As reported in **Table 3 of our paper**, our model achieves SOTA performance under these challenging conditions, demonstrating its robustness beyond standard perturbations.
>
> > Q3: In TOP-Fusion, using CLIP's text tower to estimate the probability distribution for semantic features might lead to unreliable results.
>
> **R:** We appreciate the reviewer for the comments. Several prior works have explored leveraging CLIP for AIGI detection [R1, R2]. Rather than capturing low-level artifact cues, **CLIP primarily assesses semantic plausibility, such as detection inconsistencies (e.g., unrealistic finger counts)**, thereby providing complementary signals for identifying synthetic content. This naturally motivates combining CLIP-based semantic features with artifact features that encode low-level forgery patterns.
>
> However, existing methods rely on naive fusion (e.g., direct concatenation), ignoring the discrepancy between semantic and artifact features and leading to attention dilution. We address this by processing TranX-Adapter, which explicitly models their interaction for more effective fusion.
>
> [R1] Zhou Z, Luo Y, Wu Y, et al. Aigi-holmes: Towards explainable and generalizable ai-generated image detection via multimodal large language models[C]. In ICCV 2025.
>
> [R2] Tan C, Tao R, Liu H, et al. C2p-clip: Injecting category common prompt in clip to enhance generalization in deepfake detection[C]. In AAAI 2025.
>
>
> > Q4: Is global transfer of artifact information to semantic features reasonable without accounting for their inherent locality?
>
> **R:** We appreciate the reviewer's point. We would like to clarify that **the regions exhibiting substantial discrepancies between semantic and artifact features are not required to be spatially aligned**. Instead, our design focuses on aligning their forgery probability distributions in the context of AIGI detection. Specifically, semantic features that indicate a high forgery probability are encouraged to align with artifact features that exhibit similar forgery-related patterns. This is motivated by the observation that artifact features often encode distinctive patterns when an image is AI-generated, and the goal of TOP-Fusion is to enable semantic features to capture and leverage such patterns. Moreover, the JS divergence serves as a soft weighting mechanism that adaptively modulates the interaction strength between semantic and artifact features.
>
> > Q5: Could use the TOP-Fusion in the Semantic $\rightarrow$ Artifact information transfer process yield even performance?
>
> **R:** We thank the reviewer for this valuable suggestion. We also apply TOP-Fusion in the Semantic $\rightarrow$ Artifact direction (i.e., All TOP-Fusion) and report the results in the table below. Compared with All X-Fusion, the performance shows a slight degradation with All TOP-Fusion.
>
> |   Method             | Chameleon $\uparrow$ |
>    | - | - |
>    | Qwen3-VL-2B    |          |           |
>    | w/ All TOP-Fusion  | 78.9      |
>    | w/ All X-Fusion    | 81.3      |
>    | **w/ TranX-Adapter**   | **82.3**      |
>    | Qwen3-VL-4B     |           |
>    | w/ All TOP-Fusion   | 80.7      |
>    | w/ All X-Fusion     | 81.6      |
>    | **w/ TranX-Adapter**     | **83.6**      |
>
>
> We observe that cross-attention (i.e., All X-Fusion) in the Semantic $\rightarrow$ Artifact pathway yields lower loss, suggesting more effective feature-level fusion. This is because **CLIP visual features encode rich inter-patch semantics, facilitating the injection of high-level semantic information into artifact features**. In contrast, transferring decision-level signals (e.g., real/fake probabilities) is less effective and may lose fine-grained information.

---

> > ### Author Rebuttal · Reviewer_Haa2 · 2026-04-04
> >
> > Thank you for the author's response.
> >
> > Regarding Q4:
> > "This is motivated by the observation that artifact features often encode distinctive patterns when an image is AI-generated, and the goal of TOP-Fusion is to enable semantic features to capture and leverage such patterns." This observation does not explain why the contrastive difference between semantic and artifact features does not need to consider spatial priors. A direct example is that if there is a difference between the semantic features of the eye region and the artifact features of the mouth region, the most intuitive reason is the regional difference rather than the nature of the forgery signal patterns.
> >
> > The other parts of the response have addressed my concerns.

---

> > > ### Author Response · Authors · 2026-04-07
> > >
> > > Dear Reviewer Haa2,
> > >
> > > Thank you for the insightful comment. We clarify that our design does not rely on spatial alignment, as the interaction is intentionally formulated in a different space and serves a different purpose.
> > >
> > > 1. **Non-local and globally distributed forgery cues.** In AIGI detection, the task is to determine whether an image is entirely generated by AI or originates from real-world data [R1, R2]. The forgery cues are typically non-local (i.e., not restricted to a single region) and globally distributed, meaning that artifact patterns (e.g., texture inconsistencies) and semantic inconsistencies (e.g., implausible structures) may appear in different, spatially disjoint regions, yet jointly support the same global prediction.
> > >
> > > 2. **Probability space formulation from spatial identity to decision contribution.** In TOP-Fusion, both semantic and artifact features are projected into a forgery probability space, where each patch is represented by its contribution to the global real/fake decision. Under this formulation, the interaction is no longer based on semantic identify (e.g., "eye" or "mouth"), but on the consistency of their predicted forgery evidence. As a result, ***a discrepancy between location A and B should not be interpreted as a spatial mismatch, but as an evidence gap between two types of features with respect to the same global hypothesis.***
> > >
> > > 3. **Global interaction with implicit sparsity constraint.** Although the interaction is defined globally, it is not unconstrained. As shownin our analysis (Fig. 6), the JS divergence distribution is highly non-uniform, where high-discrepancy regions are sparse and dominate the transport process. This introduces an implicit sparsity that focuses the fusion on a limited set of informative regions, preventing indiscriminate or noisy cross-region interactions.
> > >
> > > 4. Additionally, we conduct an experiment where artifact and semantic features are spatially aligned, i.e., discrepancies are computed only within the same spatial regions (w/ Aligned Spatial). As shown in the table below, the w/o Aligned Spatial (i.e., our TOP-Fusion) achieves better performance. This result indicates that the global formulation enables semantic features to absorb more comprehensive forgery evidence from artifact features, which is consistent with the non-local nature of AIGI detection.
> > >
> > >    |                                        | GenImage $\uparrow$|
> > >    | -| - |
> > >    | Qwen3-VL 2B                            |          |
> > >    | +TranX-Adapter w/ Aligned Spatial      | 82.8     |
> > >    | **+TranX-Adapter w/o Aligned Spatial** | **88.0** |
> > >    | Qwen3-VL 4B                            |          |
> > >    | +TranX-Adapter w/ Aligned Spatial      | 83.4     |
> > >    | **+TranX-Adapter w/o Aligned Spatial** | **89.8** |
> > >
> > > In summary, our method does not aim to align spatially corresponding regions, but instead leverages discrepancy as a global, task-driven signal of missing forgery evidence across artifact and semantic features, which are complementary in capturing different aspects of forgery cues and jointly support the global prediction. This design is well aligned with the non-local nature of AIGI detection.
> > >
> > > [R1] Zhou Z, Luo Y, Wu Y, et al. Aigi-holmes: Towards explainable and generalizable ai-generated image detection via multimodal large language models[C]. In ICCV 2025.
> > >
> > > [R2] Yan S, Li O, Cai J, et al. A Sanity Check for AI-generated Image Detection[C]. In ICLR 2025.

---

### Official Review · Reviewer_XHnq · 2026-03-08

**Soundness:** 3
**Presentation:** 3
**Significance:** 3
**Originality:** 3
**Overall Recommendation:** 4
**Confidence:** 4

**Summary:**

This paper addresses the problem of poor fusion performance between artifact features  and semantic features in Multimodal Large Language Models (MLLM). The authors propose a lightweight fusion adapter, TranX-Adapter, which consists of two modules: (1) Task-Aware Optimal Transfer Fusion (TOP-Fusion) and (2) X-Fusion. Experiments on three benchmarks show that this method delivers consistent performance improvements across various MLLM models.

**Compliance With Llm Reviewing Policy:**

Affirmed.

**Final Justification:**

I understand that W1 (limited novelty of individual components) is difficult to fully resolve in a rebuttal, and the argument that the contribution lies in the unified design rather than isolated modules is reasonable. That said, it does not entirely change my original assessment on this point.

Overall, I will maintain my original score.

**Key Questions For Authors:**

Q1: If the prediction head calibration is poor, will the JS-based cost matrix degrade?

Q2: What about training/inference time or GPU memory cost? Providing such information would further strengthen the paper.

**Limitations:**

No, the paper doesn't discuss the limitations.

**Strengths And Weaknesses:**

Strengths:
 1. The problem was clearly and well identified, and the motivation was compelling.
 2. Method design logic is consistent.
 3. Comprehensive experimental coverage.

 Weaknesses:
 1. The novelty of individual technical components is limited. The core novelty of propsed methods lies mainly in the design of specific application scenarios and task-aware cost matrices. The X-Fusion module is essentially a standard cross-attention module, lacking significant technological innovation.

2. The test scope of the artifact encoder is limited. Although the generalization to the VAE encoder is verified in the appendix, the main experiment relies primarily on NPR. Given that the core claim concerns the generalization of artifact feature similarity, testing on a wider range of artifact extractors would be more convincing.

---

> ### Author Rebuttal · Authors · 2026-03-31
>
> > Q1: The novelty of individual technical components is limited.
>
> **R:** We'd like to thank the reviewer for the comment. We would like to clarify that our contribution lies not in introducing a new primitive, but in identifying a previously overlooked failure mode in MLLM-based AIGI detection and designing a principled solution tailored to this issue.
>
> Specifically, our analysis reveals that **naive fusion leads to attention dilution due to the high intra-feature similarity of artifact features**, which fundamentally limits effective interaction between semantic and artifact features. This observation motivates a principled redesign of feature interaction, rather than directly reusing standard modules.
>
> In this context, X-Fusion is not a vanilla cross-attention layer. It is carefully positioned and constrained within a lightweight adapter to model the Semantic $\rightarrow$ Artifact interaction, based on the empirical finding that **cross-feature interactions predominantly occur in the shallow layers of the MLLM**. More importantly, it operates in conjunction with TOP-Fusion, which departs from standard attention by introducing a task-aware cost matrix derived from prediction discrepancies, enabling Artifact $\rightarrow$ Semantic transfer in a way that explicitly mitigates attention dilution.
>
> Therefore, the novelty of our method lies in a **unified, task-aware design of bidirectional feature interaction**, rather than in isolated architectural components. We believe this perspective provides a more principled foundation for integrating heterogeneous features within MLLMs and extends beyond a straightforward application of standard cross-attention.
>
>
> > Q2: The test scope of the artifact encoder is limited.
>
> **R:** We thank the reviewer for the suggestion. We additionally evaluate FreqNet [R1] as an alternative artifact encoder. We train on GenImage SDv1.4 and evaluate on Chameleon. The results show that TranX-Adapter consistently improves performance on both Qwen3-VL 2B and 4B. This further demonstrates **the generalization of our method beyond NPR and VAE-based artifact encoders**. We will include these results in the revised version.
>
> |     Method                       | Chameleon $\uparrow$ |
>    | - | - |
>    | Qwen3VL-2B w/o TranxAdapter |    68.5       |
>    | Qwen3VL-2B w/ TranxAdapter (FreqNet) |    73.8       |
>    | Qwen3VL-4B w/o TranXAdapter |     69.4      |
>    | **Qwen3VL-4B w/ TranXAdapter (FreqNet)** |     **74.1**      |
>
> [R1] Tan C, Zhao Y, Wei S, et al. Frequency-aware deepfake detection: Improving generalizability through frequency space domain learning[C]. In AAAI 2024.
>
> > Q3: If the prediction head calibration is poor, will the JS-based cost matrix degrade?
>
> **R:** We appreciate the reviewer for the comments. The prediction heads are either **pretrained or initialized from pretrained artifacut encoders**, ensuring reasonable calibration. They are also **jointly optimized during training in our pipeline**. Therefore, in practice, poor calibration is unlikely, and the JS-based cost matrix remains stable and effective.
>
> > Q4: What about training/inference time or GPU memory cost?
>
> **R:** We thank the reviewer for the constructive advice. To provide a clearer understanding of computational efficiency, we include a detailed comparison of training/inference time and GPU memory usage. Specifically, we conduct a controlled study on LLaVA-v1.6-mistral 7B, comparing LoRA and our TranX-Adapter under the same number of trainable parameters. As shown in the table, we report **GPU memory (GB)**, **throughput (samples per second)**, **trainable parameters (M)**, and **accuracy (%)** on Chameleon, while keeping the batch size fixed at 24 to ensure a fair comparison.
>
> The results demonstrate that TranX-Adapter introduces only a modest increase in training memory (45.6 $\rightarrow$ 51.1 GB) with no additional inference memory overhead, while maintaining comparable inference throughput. Notably, despite this minimal overhead, TranX-Adapter achieves a substantial accuracy gain of +4.7% over LoRA. These findings highlight the favorable efficiency-performance trade-off of our method, validating its practical applicability. We will include these results in the revised version.
>
>
> |    Method           | GPU Memory (Training/Inference)$\downarrow$ | Throughput (Training/Inference)$\uparrow$ | Trainable Params | Acc.$\uparrow$ |
> | - | - | - | - | -|
> | LoRA          | **45.6**/30                         | 0.04/1.2               | 40M              | 69.1 |
> | TranX-Adapter | 51.1/30                         | **0.09**/1.2                        | 40M              | **73.8** |

---

> > ### Author Rebuttal · Reviewer_XHnq · 2026-04-02
> >
> > Thank you for the rebuttal.
> >
> > The additional FreqNet experiments (W2), the efficiency comparison (Q2), and the clarification on JS-based cost matrix calibration (Q1) are all appreciated and address those concerns adequately.
> >
> > I understand that W1 (limited novelty of individual components) is difficult to fully resolve in a rebuttal, and the argument that the contribution lies in the unified design rather than isolated modules is reasonable. That said, it does not entirely change my original assessment on this point.
> >
> > Overall, I will maintain my original score.

---

> > > ### Author Response · Authors · 2026-04-07
> > >
> > > Dear Reviewer XHnq,
> > >
> > > Thank you for the thoughtful follow-up.
> > >
> > > Our key contribution is a diagnosis-driven design. We identify a previously underexplored failure mode in artifact-semantic fusion: **artifact features are highly homogeneous, which leads to nearly uniform attention when used as keys, causing attention dilution and ineffective information transfer.** This explains why naive fusion is suboptimal.
> > >
> > > Based on this, we propose TOP-Fusion to replace standard attention in the Artifact $\rightarrow$ Semantic direction. Instead of relying on feature similarity, it performs discrepancy-aware transfer in a task-relevant prediction space, which directly **mitigates attention dilution and enables more effective interaction**.
> > >
> > > For the reverse direction, we adopt X-Fusion as a **lightweight mechanism for semantic injection without modifying the LLM**. It operates as an external adapter, preserving the LLM architecture and avoiding full-model fine-tuning, thereby enabling efficient fusion with minimal additional parameters. The novelty lies not in the operator itself, but in the decomposition of interaction directions and the analysis-driven design.
> > >
> > > Thank you again for your careful reading and constructive comments.

---

### Official Review · Reviewer_7gJ3 · 2026-03-10

**Soundness:** 3
**Presentation:** 3
**Significance:** 3
**Originality:** 3
**Overall Recommendation:** 3
**Confidence:** 3

**Summary:**

This paper studies an important problem in AI-generated image detection: how to better fuse artifact features and semantic features within multimodal large language models. The authors argue that existing approaches typically concatenate the two feature types and feed them into the LLM, but artifact features exhibit high intra-feature similarity, which causes near-uniform attention in the Artifact→Semantic direction and leads to attention dilution. As a result, fine-grained artifact cues are not effectively injected into semantic representations. To address this issue, the paper proposes TranX-Adapter, a lightweight bidirectional fusion module inserted before the LLM. It includes: (1) TOP-Fusion, which constructs patch-wise discrepancy in a task-related prediction space and uses optimal transport to transfer artifact information into semantic features; and (2) X-Fusion, which injects semantic information back into artifact features via cross-attention. Experiments on multiple AIGI detection benchmarks and multiple MLLM backbones show consistent improvements and good cross-model and cross-dataset generalization.

**Compliance With Llm Reviewing Policy:**

Affirmed.

**Final Justification:**

While the authors' rebuttal has addressed some of my concerns, I remain concerned that the core contribution still relies heavily on integrating established components without providing deeper analytical justification. Therefore, I maintain my original rating. I appreciate the authors' efforts and hope they can strengthen the manuscript.

**Key Questions For Authors:**

1.The paper attributes the gains mainly to alleviating attention dilution. Can the authors provide stronger causal evidence, beyond correlational analysis, for this claim?
2.Why is JS divergence the best choice for the transport cost? Beyond COS and KL, did the authors explore learned metrics or uncertainty-aware discrepancy designs?
3.How much of the gain comes from the task-aware transport design itself, versus simply adding another trainable fusion module?
4.Do the authors expect TranX-Adapter to generalize to other tasks that require low-level artifact cues and high-level semantics to be jointly modeled? If so, what are the most natural target tasks?

**Limitations:**

The impact statement appropriately acknowledges that such detection tools could be misused adversarially and should support rather than replace human verification.
However, the discussion of technical limitations could be stronger, especially regarding dependence on artifact encoder quality, behavior under severe post-processing, and robustness boundaries under more challenging generation distributions.

**Strengths And Weaknesses:**

A major strength of the paper is that it identifies a concrete failure mode rather than proposing a fusion module in isolation. The authors first diagnose that the homogeneity of artifact features weakens effective feature interaction inside the LLM, making naive concatenation suboptimal. This diagnosis gives the method a strong motivation. The design is also targeted: instead of using a single symmetric fusion strategy, the paper uses TOP-Fusion for the harder Artifact→Semantic path and X-Fusion for the reverse Semantic→Artifact path. This asymmetric design is sensible. The empirical section is reasonably strong, including cross-model and cross-dataset evaluations, ablations, cost-matrix comparisons, and PEFT comparisons.
The main weaknesses are as follows. First, the novelty is moderate: the contribution is best characterized as a combination of a clear diagnosis and a targeted fusion design, rather than a broadly new learning principle. Second, while the empirical justification for using JS divergence in TOP-Fusion is reasonable, the theoretical rationale remains limited. Third, the claims are mainly validated within AIGI detection, and the broader applicability of the method to other artifact-semantic fusion problems is not established. Fourth, although the experiments are already fairly complete, the conclusions would be stronger with more comparisons against stronger modern fusion baselines such as gated routing, token mixers, or more adaptive fusion strategies.

---

> ### Author Rebuttal · Authors · 2026-03-31
>
> > Q1: The contribution is a diagnosis and a fusion design, rather than a broadly new learning principle.
>
> **R:** We appreciate the reviewer's comments. While our work is indeed motivated by a clear diagnosis of existing limitations, it goes beyond a straightforward combination by introducing a principled perspective on semantic-artifact interaction in MLLMs.
>
> Although recent efforts have incorporated MLLMs into AIGI detection to enable both classification and explanation, and prior studies have shown the benefits of leveraging artifact features, existing approaches largely rely on naive feature concatenation. This design fails to address the fundamental issue in feature interaction, **where the high internal similarity of artifact features leads to indistinguishable key representations in self-attention, resulting in nearly uniform attention distributions and diminished interaction effectiveness**.
>
> In contrast, we explicitly model their bidirectional interactions, accounting for their distinct characteristics. Based on this insight, we propose TranX-Adapter, which is empirically validated and consistently yields more effective feature interaction.
>
> Furthermore, we demonstrate that **our method extends beyond classification to support AIGI detection with explanation (see Reviewer Sfo2, Q1)**, highlighting its robustness and broader applicability.
>
> > Q2: Why is JS divergence the best choice for the transport cost?
>
> **R:** We'd like to thank the reviewer for the comments. The goal of TOP-Fusion is to capture task-relevant inconsistencies between the two feature types in the probability space, **rather than relying on a specially designed discrepancy metric**. To validate that our method does not depended on a particular metric, we compare COS, KL and JS divergences and observe consistent improvements across all (Table 8), confirming that our method is not tied to a specific metric, though JS performs best due to its **symmetric and stable properties**. Unlike COS, which measures feature similarity in embedding space, our TOP-Fusion operates on prediction distributions, making divergence-based more appropriate; thus, the gains mainly stem from our discrepancy-aware transport formulation rather than the choice of metric.
>
>
> > Q3: How much of the improvement is due to the task-aware transport design versus simply adding a trainable fusion module?
>
> **R:** We thank the reviewer for the suggestion. We include an additional baseline based on GMU [R1], where a learnable gating network predicts fusion weights to combine artifact and semantic features. As shown in the table below, TranX-Adapter consistently outperforms GMU on both GenImage and Chameleon, indicating that **the gain is not simply from adding a trainable fusion module**. TOP-Fusion models the discrepancy between artifact and semantic prediction distributions, enabling fine-grained and task-relevant alignment.
>
> |  Method     | GenImage (Trained on SDv1.4) $\uparrow$ | Chameleon (Tranined on GenImage) $\uparrow$ |
>    | - | - | - |
>    | Qwen3-VL-2B   |              |                                  |
>    | w/ GMU           | 84.2         | 79.7                             |
>    | **w/ TOP-Fusion** | **88.0**          | **82.3**                             |
>    | Qwen3-VL-4B   |               |                                  |
>    | w/ GMU           | 86.7          | 79.9                             |
>    | **w/ TOP-Fusion** | **89.8**          | **83.6**                             |
>
>
> [R1] Arevalo J, Solorio T, Montes-y-Gómez M, et al. Gated multimodal units for information fusion[J]. arXiv preprint arXiv:1702.01992, 2017.
>
>
> > Q4: Can TranX-Adapter generalize to tasks requiring joint modeling of low-level artifacts and high-level semantics?
>
> **R:** Yes, we expect TranX-Adapter to generalize beyond AIGI detection. Its design addresses a challenge in fusion heterogeneous features within distinct distributions. It can be naturally extended to tasks that require combining low-level artifact cues and high-level semantics, such as **medical imaging analysis** and **industrial inspection** where subtle artifact features must be integrated with global semantic understanding.
>
> > Q5: Lacking causal evidence, beyond correlational analysis.
>
> **R:** We'd like to thank the reviewer for the comments. We provide a causal explanation grounded in the self-attention mechanism. When artifact features serve as keys, their high intra-feature similarity renders different keys nearly indistinguishable (Figure 4 (a)). Consequently, queries assign nearly uniform attention weights across positions, resulting in attention dilution (Figure 4 (c)), which hinders the model from focusing on informative regions.
>
> This mechanism is further supported by empirical evidence: alleviating such uniform attention leads to stronger information flow and consistent performance gains (Sec. 5.5), indicating that **mitigating attention dilution is a primary driver of the observed improvements**.

---

> > ### Author Rebuttal · Reviewer_7gJ3 · 2026-04-03
> >
> > Thanks for the detailed response.
> > However, I remain concerned that the core contribution still relies heavily on integrating established components without providing deeper analytical justification. Therefore, I maintain my original rating. I appreciate the authors' efforts and hope they can strengthen the manuscript.

---

> > > ### Author Response · Authors · 2026-04-07
> > >
> > > Dear Reviewer 7gJ3,
> > >
> > > Thank you for the thoughtful follow-up. We would like to clarify that the core contribution of our work is **NOT** a simple combination of existing modules, but a diagnosis-driven fusion design grounded in the specific interaction failure between artifact and semantic features within MLLMs.
> > >
> > > 1. **Our contribution begins with identifying a previously underexplored failure mode in artifact-semantic fusion.** Our analysis shows that directly fusing artifact features with semantic features inside the MLLM leads to attention dilution. The key reason is that artifact features are substantially more homogeneous and less discriminative in feature space than semantic features, which causes the attention map to become nearly uniform when artifact features serve as keys in the Artifact $\rightarrow$ Semantic interaction. This weakens the transfer of informative forgery cues rather than enabling effective fusion. We believe this diagnosis is central, because it explains why naive concatenation is suboptimal in this setting, rather than merely observing that a new module performs better. This motivation is explicitly reflected in the paper's pilot study and attention analysis (Figure 4).
> > >
> > > 2. **TOP-Fusion is proposed specifically to address this failure mode, rather than as a generic add-on module.** Based on the above finding, we design TOP-Fusion to replace cross-attention interaction in the Artifact $\rightarrow$ Semantic direction with a discrepancy-aware transfer mechanism. Instead of relying on cross-attention over highly similar artifact features, TOP-Fusion maps both artifact and semantic features into a task-relevant prediction space and uses the JS divergence between their prediction distributions as the transport cost. In this way, the fusion process is guided toward regions where the two feature types disagree most strongly in terms of forgery evidence, which directly mitigates the attention-dilution problem. Therefore, the novelty is not the isolated use of optimal transport itself, but the observation that artifact-semantic fusion in MLLMs requires a different interaction principle from standard attention, together with a concrete formulation that resolves this issue.
> > >
> > > 3. **The proposed TOP-Fusion leads to more effective cross-feature interaction.** We further provide evidence that this design changes the fusion behavior in a meaningful way. As shown in the table below, TOP-Fusion significantly enhances the Artifact $\rightarrow$ Semantic interaction, leading to substantial performance improvements in the AIGI detection task. As illustrated in Figure 6 (our paper), TOP-Fusion highlights regions with pronounced discrepancies between artifact and semantic features, while also achieving lower training loss.
> > >
> > >    |                 | Information flow $S_{artifact\rightarrow semantic} \uparrow$ | Chameleon $\uparrow$ |
> > >    | --------------- | --------------------------------------------------- | --------- |
> > >    | Cross Attention | 0.12                                                | 81.9      |
> > >    | Our TOP-Fusion  | **0.30**                                               | **85.1**      |
> > >
> > >
> > >
> > > 4. **X-Fusion is a lightweight and efficient mechanism for semantic $\rightarrow$ artifact interaction without modifying the LLM.**  Our analysis shows that ***visual-feature interaction in the LLM mainly emerges in shallow layers, suggesting that effective artifact-semantic interaction can be handled before the LLM by a lightweight module, without modifying or fine-tuning the full language model.*** This is why we adopt X-Fusion for the Semantic $\rightarrow$ Artifact direction: it provides a simple and parameter-efficient mechanism for semantic injection while preserving the LLM architecture and avoiding unnecessary full-model adaption. In this sense, the novelty lies in the diagnosis-guided decomposition of the two interaction directions and the corresponding fusion strategy, rather than in claiming novelty for cross-attention alone.
> > >
> > > Thank you again for your careful reading and constructive comments.

---

### Official Review · Reviewer_Sfo2 · 2026-03-11

**Soundness:** 3
**Presentation:** 3
**Significance:** 3
**Originality:** 3
**Overall Recommendation:** 4
**Confidence:** 4

**Summary:**

This paper tackles the problem of attention dilution and poor feature fusion performance caused by the high intra-feature similarity of artifact features when MLLMs integrate artifact features and semantic features for AIGI detection. It proposes a lightweight TranX-Adapter, which realizes the effective transfer of artifact features to semantic features through TOP-Fusion, and then achieves the reverse injection of semantic features into artifact features via X-Fusion, thus completing the bidirectional and efficient fusion of the two types of features. The contributions of this paper lie in revealing the attention dilution problem of MLLMs in AIGI detection tasks, designing a lightweight and efficient bidirectional feature fusion mechanism, verifying the improvement of detection performance and robustness of this method on different MLLMs through experiments, and meanwhile, this method only fine-tunes the adapter parameters, which improves the training efficiency.

**Compliance With Llm Reviewing Policy:**

Affirmed.

**Final Justification:**

The rebuttal changed my evaluation. The supplementary experiments and analyses resolved my primary concerns. I am raising my overall recommendation from 3 to 4.

**Key Questions For Authors:**

1.This paper only limits the task to the binary classification of authentic and fake AIGIs, and does not carry out research on the interpretability of detection by taking advantage of the semantic understanding and reasoning capabilities of MLLMs. Have the authors attempted to implement such interpretability reasoning based on TranX-Adapter? If yes, are there any technical difficulties or will it lead to a significant loss of detection performance? If no, what are the core considerations?

2.This paper only verifies the effect of TranX-Adapter on MLLMs such as LLaVA-1.6-mistral 7B and Qwen3-VL 2B/4B, and does not set up comparative experiments with traditional small-model detectors. Can such experiments be supplemented to show the performance comparison between MLLMs combined with TranX-Adapter and traditional small-model detectors? In addition, does TranX-Adapter have the technical foundation for adaptation to traditional small models? Can relevant experiments be supplemented to verify the adaptability of TranX-Adapter to traditional small models?

3.The three datasets GenImage, Chameleon and RRDataset adopted in the experiments have significant overlaps in core tasks, generator sources and data scenarios. Have the authors conducted experiments on TranX-Adapter with more datasets or in other diversified scenarios? Can relevant experiments be considered to be supplemented to verify the robustness and generalization ability of TranX-Adapter in more scenarios?

After addressing the questions above, I will consider raising the Soundness and Presentation scores to 3.

**Limitations:**

No. The authors do not set up an independent Limitations section in the main text to elaborate on the shortcomings of their research in terms of theory, technology and experimental design, and only mention a small number of potential social and ethical risks in the Impact Statements. It is recommended to supplement the analysis of the limitations of this method at the technical and experimental levels, such as its adaptability to traditional small models, the expandability of task scenarios, and the support of dataset diversity.

**Strengths And Weaknesses:**

**Strengths**


1.The author accurately identifies the attention dilution problem caused by the high intra-feature similarity of artifact features, and designs a bidirectional feature fusion mechanism for the two types of features based on TOP-Fusion and X-Fusion as a targeted solution, with a self-consistent technical path logic.

2.It explicitly reveals the inherent correlation between the attention dilution problem and the high intra-feature similarity of artifact features for the first time, and verifies this conclusion through both quantitative indicators and visualization results, providing a new research perspective and direction for the subsequent research of MLLMs in the field of AIGI detection.

**Weaknesses**


1.The paper limits the task to the binary classification of authentic and fake AIGIs, and fails to conduct research on the interpretability reasoning of detection results by leveraging the semantic understanding and reasoning advantages of MLLMs, thus not giving full play to the inherent capabilities of MLLMs in multimodal tasks.

2.With the research only focusing on the relatively simple task setting of binary classification of authentic and fake AIGIs, the paper only demonstrates the performance improvement effect of MLLMs with small and medium parameter scales such as LLaVA-1.6-mistral 7B and Qwen3-VL 2B/4B, and does not set up comparative experiments with traditional small-model detectors. This makes it impossible to verify whether the performance of MLLMs combined with TranX-Adapter is superior to that of traditional small-model detectors, as well as the adaptability of TranX-Adapter to traditional small models.

3.The experiments only adopt three datasets, namely GenImage, Chameleon and RRDataset, and there is a certain overlap in the test scenarios and data distributions of various datasets. The data support dimension of the experimental conclusions is insufficient, which is not enough to fully prove the robustness and generalization ability of the method in a wider range of data scenarios.

---

> ### Author Rebuttal · Authors · 2026-03-31
>
> > Q1: The paper restricts the task to binary classification and fails to conduct research on the interpretability reasoning of detection results.
>
> **R:** We appreciate the reviewer's comments. In this work, our primary objective is to investigate how to enable more effective fusion between artifact and semantic features within MLLMs. To this end, we deliberately formulate the task in its most fundamental form for AIGI detection, i.e., binary classification based solely on the input image and a prompt. This simplified setting allows us to isolate and analyze the impact of feature fusion without introducing additional complexities from more advanced reasoning tasks.
>
> Importantly, our TranX-Adapter is not limited to pure classification. Following AIGI-Holmes [R1], we incorporate supervised fine-tuning with both **labels** and **textual rationales**, enabling the model to produce interpretable explanations alongside predictions. As shown in the table below, our TranX-Adapter achieves consistently stronger performance than AIGI-Holmes, demonstrating that **the effectiveness of our fusion design for AIGI detection tasks.**
>
>
>    | Method | Chameleon $\uparrow$ | AIGCDetection $\uparrow$ |
>    | - | - | -|
>    | AIGI-Holmes  | 75.9      | 93.2   |
>    | **LLaVA-1.6-mistral 7B w/ TranX-Adapter** | **87.5**  | **95.3**  |
>
> [R1] Zhou Z, Luo Y, Wu Y, et al. Aigi-holmes: Towards explainable and generalizable ai-generated image detection via multimodal large language models[C]. In ICCV 2025.
>
> > Q2: Lacking comparisons to traditional detectors, the study cannot validate TranX-Adapter's superiority or generalizability.
>
> **R:** We'd like to thank the reviewer for the comments. Our comparisons in **Tables 1, 2 and 3 of our paper already include traditional small-model detectors (e.g., F3Net [R2], DIRE [R3] and UnivFD [R4])**, where our method achieves SOTA performance. Furthermore, to explicitly evaluate the adaptability of TranX-Adapter to traditional small models, we integrate it into CO-SPY [R5], a small-model detector, by replacing its original adaptive fusion module. The model is trained on ImageNet real images and SDv1.4-generated images, and evaluated on GenImage. As shown in the table below, this substitution yeilds a **+4.5%** accuracy improvement, **demonstrating the effectiveness of our TranX-Adapter on traditional small models**.
>
> | Method | GenImage $\uparrow$ |
> | - | - |
> | F3Net  |  68.7    |
> | DIRE                    |  70.7    |
> |UnivFD                   |   73.3   |
> | CO-SPY                  | 76.5     |
> | **CO-SPY w/ TranX-Adapter** | **81.0**     |
>
>
> [R2] Qian Y, Yin G, Sheng L, et al. Thinking in frequency: Face forgery detection by mining frequency-aware clues[C]. In ECCV 2020.
>
> [R3] Wang Z, Bao J, Zhou W, et al. Dire for diffusion-generated image detection[C]. In ICCV 2023.
>
> [R4] Ojha U, Li Y, Lee Y J. Towards universal fake image detectors that generalize across generative models[C]. In CVPR 2023.
>
> [R5] Cheng S, Lyu L, Wang Z, et al. Co-spy: Combining semantic and pixel features to detect synthetic images by ai[C]. In CVPR 2025.
>
> > Q3: The experiments only adopt three datasets.
>
> **R:** We appreciate the reviewer's comments. We follow the experimental protocol of AIDE [R6], which also adopts three benchmarks for evaluation. Specifically, GenImage is one of the most commonly used benchmarks for AIGI detection, designed to evaluate cross-generator generalization. Chameleon focuses on samples that are genuinely challenging even for human perception, thereby emphasizing fine-grained semantic and perceptual discrimination. RRDataset covers a diverse range of real-world scenarios, including robustness to Internet transmission and re-digitization processes, which are critical for practical deployment. To further strengthen the empirical evidence, we additionally conduct experiments on two benchmarks: **AIGCDetection [R7] and AIGIBench [R8]**, which introduce broader data distributions and more diverse generation sources. As shown in the table below, **our method consistently achieves SOTA performance compared to existing approaches**.
>
> |    Method        | AIGCDetection $\uparrow$ | AIGIBench $\uparrow$ |
>    | - | - | - |
>    | NPR                |        82.9      |     67.9     |
>    | AIDE               |        92.8       |      77.6     |
>    | AIGI-Holmes        |        93.2           |      77.4           |
>    | Qwen3-VL-2B  w/ TranX-Adapter      | 89.8          | 81.5      |
>    | Qwen3-VL-4B   w/ TranX-Adapter     | 88.5          | 81.6      |
>    | **LLaVA-v1.6-mistral 7B w/ TranX-Adapter** |  **93.5**          | **83.4**      |
>
>
> [R6] Yan S, Li O, Cai J, et al. A Sanity Check for AI-generated Image Detection[C]. In ICLR 2024.
>
> [R7] Zhong N, Xu Y, Li S, et al. Patchcraft: Exploring texture patch for efficient ai-generated image detection[J]. arXiv preprint arXiv:2311.12397, 2023.
>
> [R8] Li Z, Yan J, He Z, et al. Is Artificial Intelligence Generated Image Detection a Solved Problem?[C]. In NeurIPS 2025.

---

> > ### Author Rebuttal · Reviewer_Sfo2 · 2026-04-04
> >
> > My concerns have been adequately addressed. I will increase my score accordingly.

---

> > > ### Author Response · Authors · 2026-04-07
> > >
> > > Dear Reviewer Sfo2,
> > >
> > > We sincerely thank the reviewer for the thoughtful and constructive feedback throughout the discussion. We are glad that our clarifications have addressed the main concerns, and we greatly appreciate the positive assessment of our work.

---

### Decision · Program_Chairs · 2026-04-30

**Decision:**

Accept (regular)

**Comment:**

Overall, The rebuttal is solid. It directly addresses the reviewers’ questions, adds new evidence, and covers several practical gaps, including small-model comparisons, extra datasets, additional artifact encoders, efficiency, and prompt robustness. The authors' rebuttal comments successfully changed one reviewer's score.

Its main limitation is that it resolves empirical concerns better than conceptual ones. In particular, it does not fully adress the criticisms that the novelty is limited, that the evidence for the attention-dilution mechanism is still more correlational than truly causal, and that the justification for global transport without spatial locality remains somewhat fragile.

In short, the rebuttal is strong enough to reassure supportive reviewers and to improve at least one score, but probably not strong enough to fully change the mind of a skeptical reviewer with deeper concerns about novelty or analytical justification. Considering the average score and author-reviewer discussion, I am inclined to acceptance for this paper.